# Optical force mapping at the single-nanometre scale

Junsuke Yamanishi [1,2], Hidemasa Yamane [3,4], Yoshitaka Naitoh [1], Yan Jun Li [1], Nobuhiko Yokoshi [3], Tatsuya Kameyama[5], Seiya Koyama[5], Tsukasa Torimoto [5], Hajime Ishihara [3,6,7 ✉] & Yasuhiro Sugawara [1 ✉]

Three-dimensional (3D) information of the optical response in the nanometre scale is important in the field of nanophotonics science. Using photoinduced force microscopy (PiFM), we can visualize the nano-scale optical field using the optical gradient force between the tip and sample. Here, we demonstrate 3D photoinduced force field visualization around a quantum dot in the single-nanometre spatial resolution with heterodyne frequency modulation technique, using which, the effect of the photothermal expansion of the tip and sample in the ultra-high vacuum condition can be avoided. The obtained 3D mapping shows the spatially localized photoinduced interaction potential and force field vectors in the single nano-scale for composite quantum dots with photocatalytic activity. Furthermore, the spatial resolution of PiFM imaging achieved is ~0.7 nm. The single-nanometer scale photoinduced field visualization is crucial for applications such as photo catalysts, optical functional devices, and optical manipulation.

[1] Department of Applied Physics, Osaka University, Suita, Osaka, Japan. [2] Institute for Molecular Science, National Institutes of Natural Sciences, Okazaki, Aichi, Japan. [3] Department of Physics and Electronics, Osaka Prefecture University, Sakai, Osaka, Japan. [4] Department of Physics, Kitasato University, Sagamihara, Kanagawa, Japan. [5] Department of Materials Chemistry, Graduate School of Engineering, Nagoya University, Chikusa-ku, Nagoya, Aichi, Japan. [6] Department of Materials Engineering Science, Osaka University, Toyonaka, Osaka, Japan. [7] Center for Quantum Information and Quantum Biology, Osaka University, Suita, Osaka, Japan. ✉email: ishi@mp.es.osaka-u.ac.jp; sugawara@ap.eng.osaka-u.ac.jp

Three-dimensional (3D) distributions of photoinduced fields at or beyond the nanometre scale[1–3] are essential for deducing the full dimensional symmetry of molecular excited states that govern the internal phenomena in specially functionalised quantum dots (QDs)[4] or molecules[5,6] and complex molecular substances[7,8]; in addition, the gradients of the 3D distributions are required in the form of vector maps for analysing and designing approach paths for optical trapping[9,10]. Therefore, aside from the theoretical speculation or ensemble measurements of individual materials, the direct visualisation of these photoinduced electric fields in individual materials is crucial for creating optical functions in the single-nanometre-scale range. However, this task is challenging because of the sensitivity bottlenecks in the current state-of-the-art visualisation technologies.

Here, we demonstrate the visualisation of photoinduced electric field distributions on composite QDs that have special electronic and optical structures as photocatalysts at the single-nanometre scale using photoinduced force microscopy (PiFM)[11]. This is realised via 3D mapping using PiFM. In particular, PiFM is used to observe the optical gradient force proportional to the gradient of the electric field intensity ($F_{grad} \propto \nabla|E|^2$)[3]. Our 3D mapping shows the spatially inhomogeneous photoinduced interaction potential and force field vectors related to the intensity of the electric field ($U_{grad} \propto |E|^2$) and field variations, respectively. Our 3D mapping for the single-nanometre-scale visualisation was realised by conducting PiFM measurements under an ultra-high vacuum (UHV). There are important benefits of performing observations under vacuum[12,13] which are due to the near absence of atmospheric molecules in such conditions. In particular, the absence of air significantly increases the cantilever force sensitivity[14] and the thermal stability of the PiFM measurements[15]. In addition, owing to the vacuum conditions, no water layer is formed on the surface of the sample, which could otherwise become problematic for the detection of dipole–dipole interactions between the tip and the sample[16]. Nevertheless, some issues do arise when PiFM measurements are conducted under vacuum. One issue is that laser modulation causes photothermal vibrations in the cantilever[12], another is that the laser modulation frequency does not shift in accordance with the shift in the resonance frequency of the cantilever. These issues result in substantial artifacts in the PiFM signal. Therefore, we propose the heterodyne frequency modulation (heterodyne-FM) technique to address such artifacts[13]. In our heterodyne-FM technique, the light intensity of the laser used for PiFM measurements is modulated with a frequency of $2f_1 + f_m$, where $f_1$ is the resonance frequency of the cantilever, and $f_m$ is the frequency selected from the bandwidth range of the phase-locked loop circuit that is used to detect the resonance frequency shift of the cantilever ($\Delta f$) via frequency-modulation atomic force microscopy (FM-AFM)[14]. In our study, $\Delta f$ was also used to control the tip–sample distance. The modulation of the photoinduced force results in $\Delta f$ being modulated as $f_m$ ($\Delta f(f_m)$). We measured the optical gradient force by detecting the modulated signal ($\Delta f(f_m)$) using a lock-in amplifier; additionally, the non-delayed lock-in X (LIX) component was measured as $\Delta f(f_m)X$ (see Methods section for more details). Because the light-intensity modulation frequency ($2f_1 + f_m$) is significantly different from the cantilever resonance frequency ($f_1$) in the proposed heterodyne-FM technique, the modulated signal ($\Delta f(f_m)$) is not affected by small photothermal vibrations of the cantilever. Moreover, the frequency of the heterodyne-FM PiFM signal ($f_1 + f_m$) shifts as the resonance frequency of the cantilever shifts. Therefore, the photoinduced force can be detected without interference from the shift. This achievement can significantly expand the study of optical and mechanical processes in various fields of research and enhance the functionalisation of optical materials at the single-nanometre-scale.

## Results

**Simultaneous PiFM measurements performed using multiple wavelengths.** To clearly demonstrate gradient force detection through our heterodyne-FM technique, we performed PiFM imaging (Fig. 1a) using Zn–Ag–In–S (ZAIS) QDs consisting of multiple different optical components[4]. The QD structure (heterostructure) has been developed using a state-of-the-art chemical synthesis technology to realise enhanced photocatalytic activity arising from the designed electronic level scheme (see Fig. 1b). The visualisation of the internal optical structures of individual QDs is crucial for establishing the designed electronic scheme and targeted catalytic activity. The laser light was incident from the side of the probe onto the ZAIS QDs and a thin gold film fabricated on a mica substrate at an angle of 70° (see the Methods section for more experimental details). To avoid accounting for laser spot accuracy in our measurements, the observed PiFM images were normalised with respect to $\Delta f(f_m)X$ on the thin gold film whose thickness was ~100 nm. The structural model of the observed ZAIS QDs is shown in Fig. 1b[4]. Each ZAIS QD has a dumbbell structure with ellipsoid crystals at both ends of a nanorod. Such a structure in these QDs introduces different electronic levels at their ends, 1.97 eV, and in the middle, 2.92 eV, respectively. In turn, this difference in the electronic levels at the ends and at the middle of the QDs leads to different molar absorption coefficient dispersion behaviours (Fig. 1c). Figure 1d is an AFM image that shows the dumbbell-structured ZAIS QDs, and Fig. 1e and f show the corresponding PiFM images simultaneously obtained using laser irradiation at 660 nm (1.88 eV) and 520 nm (2.38 eV) (see the Methods section for details regarding this multiple- (two-) wavelength measurement). In particular, in Fig. 1e, a strong photoinduced force ($\Delta f(f_m)X$) can be observed clearly; this force is localised at the dumbbell-shaped ends of the ZAIS QDs (nanoellipsoids). Furthermore, as can be seen from Fig. 1f, the photoinduced force in the middle part of the ZAIS QD (nanorod) is similar to that on the nanoellipsoids. This difference between Fig. 1e and f can be observed from the line profiles of the abovementioned forces (Fig. 1g); in particular, these line profiles indicate that the ratio of the photoinduced force on the nanoellipsoids to that on the nanorod is ~1.88 and ~1.20 at 660 nm and 520 nm, respectively. Moreover, images (Fig. 1l and m) and line profiles (Fig. 1o) of the ZAIS QD obtained by theoretical calculations using the discrete dipole approximation (DDA) method (see the Theoretical method section in Supplementary Note 11 for details on the DDA method) indicate this feature with similar ratios of ~1.59 and ~1.07 at 660 nm and 520 nm, respectively. A similar set of experimental images obtained using two different wavelengths is shown in Fig. 1h–j. In particular, Fig. 1h shows the acquired AFM image and Fig. 1i and j show the associated PiFM images acquired at 660 nm and 785 nm (1.58 eV), respectively. The PiFM image acquired at 660 nm (Fig. 1i) shows a strong photoinduced force at the nanoellipsoids, which is similar to that in Fig. 1e. Moreover, similar to the 660 nm case (Fig. 1i), the PiFM image acquired at 785 nm (Fig. 1j) also reveals a strong photoinduced force on the nanoellipsoids. In addition, this phenomenon is also clear from the corresponding line profiles (Fig. 1k). These strong photoinduced forces were also reproduced via theoretical calculations for the case depicted in Fig. 1l and n as well as the corresponding line profile shown in Fig. 1o. This similarity between the theoretical calculations and PiFM measurements confirms that optical spectromicroscopy can be performed at the nanoscale by detecting gradient forces. With regard to this measurement, one may fear that the strengths of the non-photoinduced forces, i.e., the electrostatic forces arising from the difference in the surface photovoltages or surface

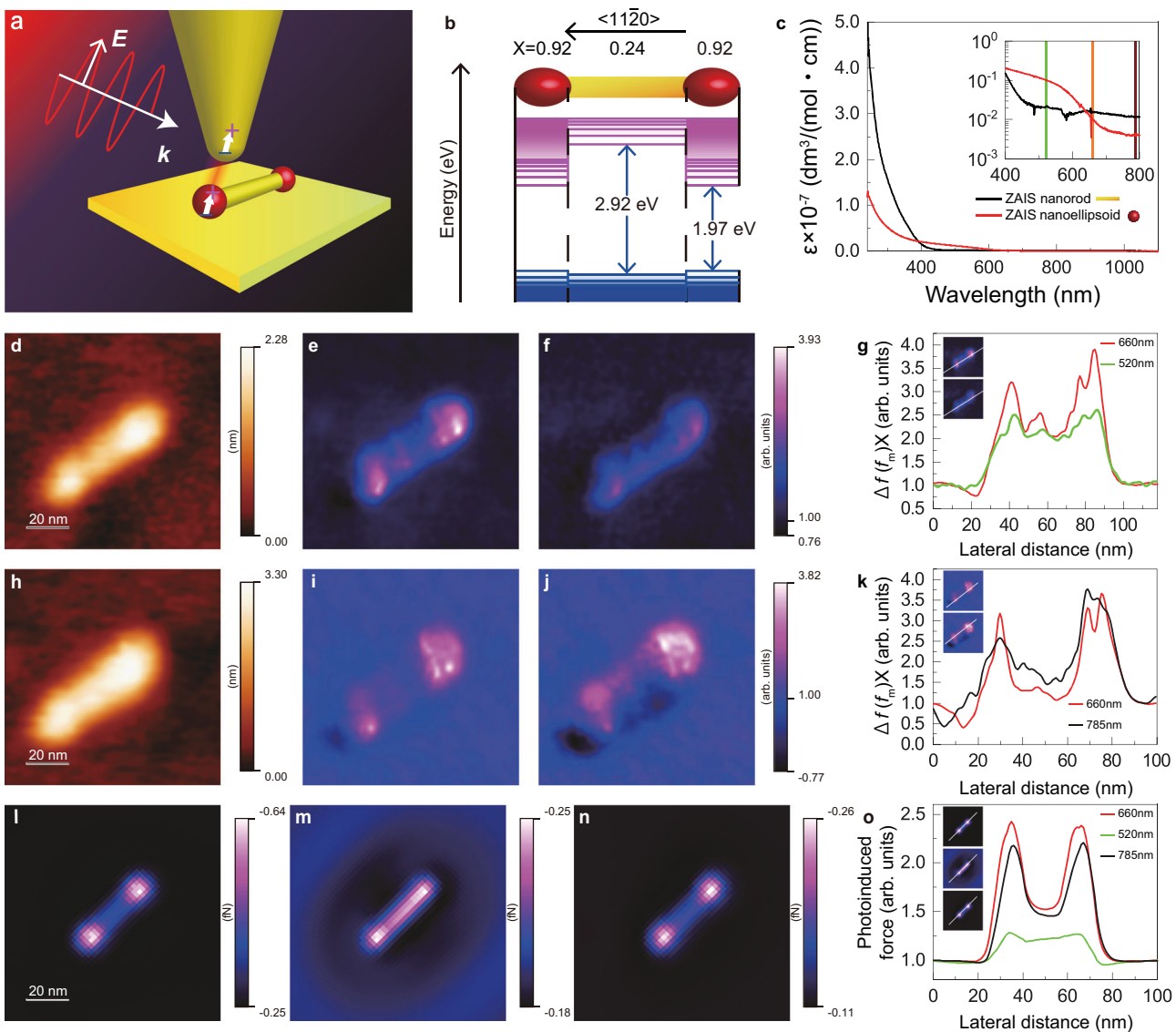

**Fig. 1 Simultaneous PiFM measurements performed using two different wavelengths. a** Schematic diagram of PiFM under UHV in a side-illumination configuration. Here, **E** and **k** are the electric field and wave number vector, respectively, of the incident light. **b** Model of a dumbbell-shaped ZAIS QD and structure of its electronic states. $x$ reflects the composition of the ZAIS QD as described by the formula $(AgIn)_xZn_{2(1-x)}S_2$. The band gaps of the nanoellipsoids and nanorod are approximately 1.97 and 2.92 eV, respectively. **c** Molar absorption coefficient ($\epsilon$) dispersion at the end and in the middle of a ZAIS QD. **d**, **h** AFM images of a ZAIS QD on a gold surface. **e**, **i** PiFM images acquired at $\lambda = 660$ nm. **f**, **j** PiFM images acquired at $\lambda = 520$ and 785 nm, respectively. **d**–**f** and **h**–**j** were acquired simultaneously. **g** Photoinduced force profiles for the images shown in **e** and **f**. **k** Photoinduced force profiles for the images shown in **i** and **j**. The positions at which the profiles were acquired are indicated in the insets. **l**–**n** Theoretically calculated photoinduced force images at $\lambda = 660$ nm, 520 nm, and 785 nm. **o** Line profiles of **l**–**n** in the insets. The photoinduced forces are normalised with respect to those on the gold surface.

potentials of the edges and rod, might vary with the photoinduced force signal itself or the distance from the tip to the sample surface, and this may, in turn, cause the strength of the gradient force signals on each site to vary, as observed in the PiFM images. However, the energy of the incident light was below the bandgap energy of the QDs, particularly when using 660 and 785 nm for both the edge and rod of the QD. The use of these energies of the light enabled us to avoid the generation of the surface photovoltages. Furthermore, in this measurement, the van der Waals force was the dominant force detected by the tip for the tip–sample distance control because the electrostatic force originating from the contact potential difference between the gold tip and the substrate was negligible. These facts can be confirmed from the nearly similar force curves shown in Supplementary

Fig. 7b in Supplementary Note 2. (For more detailed discussions of the surface photo-voltage, see the Methods section and the influence of surface photo-voltage section in Supplementary Note 7.)

**High resolution PiFM imaging.** To depict the small variations on the edges of the QD nanoellipsoids in more detail, we present the AFM and PiFM images from a narrower scan range than in the previous cases in Fig. 2. In particular, Fig. 2a shows the acquired AFM image with the narrower scan range, while Figs. 2b and c show the PiFM images acquired using the laser wavelengths of 660 and 785 nm, respectively. The nanoellipsoid and nanorod components in the AFM image are observed as having elliptical

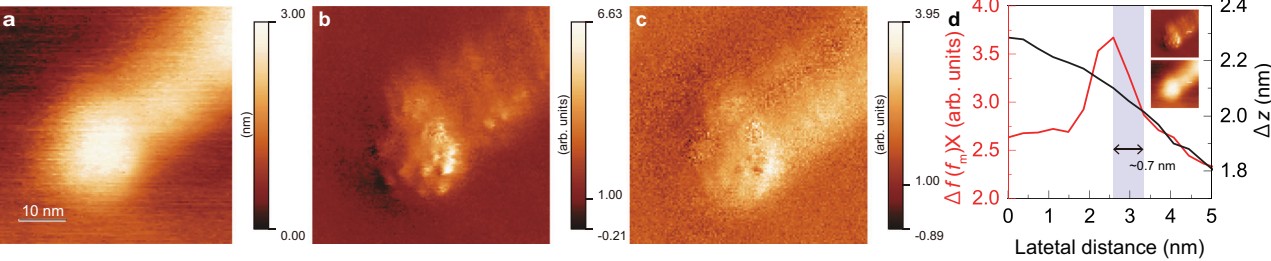

**Fig. 2 PiFM imaging on one nanoellipsoid of a ZAIS QD. a** AFM image of the ZAIS QD. **b**, **c** PiFM images of the ZAIS QD at 660 and 785 nm, respectively. **d** $\Delta f(f_m)X$ at 660 nm, and the profile of $\Delta z$ in the middle of the ZAIS QD. This plot indicates that a spatial imaging resolution of ~0.7 nm was achieved. The positions at which the profiles were acquired are indicated in the insets. During imaging, the tip–sample distance was controlled via feedback control ($\Delta f = -20$ Hz) with $A = 10$ nm.

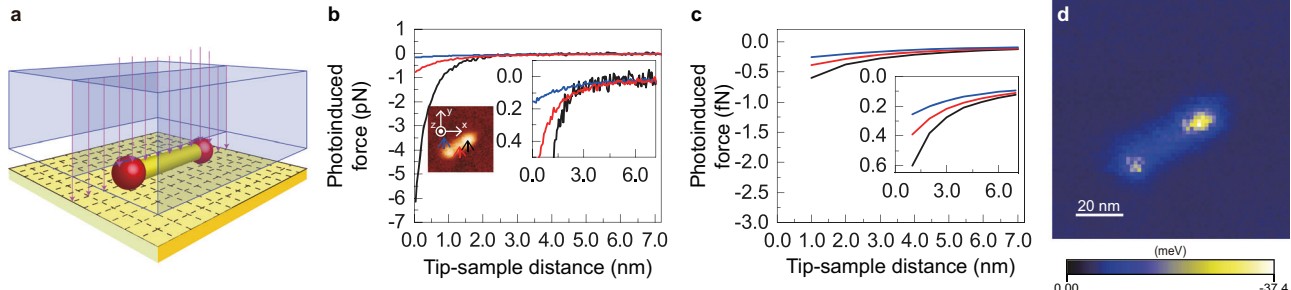

**Fig. 3 3D force field mapping of the photoinduced force. a** Model of the ZAIS QD on gold during the measurement. **b** Force curves ($Fz_{pif} - \Delta z$) of the photoinduced force at the end of ZAIS QD, in the middle of the QD, and on the gold film as recorded in the mapping with a controlled tip–sample distance ($\Delta f = -28$ Hz, $A = 10$ nm, and $\lambda = 660$ nm). The inset image indicate the sites where the force curves are measured. **c** Theoretically calculated force curves for the photoinduced force. The force curves shown in the insets in **b** and **c** are the zoomed-in version of the graphs shown in the subfigure. **d** Interaction potential at $\Delta z = 0.0$ nm.

and cylindrical in shapes, respectively. However, the internal structures of these components cannot be visualised. In contrast, in the corresponding PiFM image (Fig. 2b), structures at a scale of approximately several nanometres are observed inside the components even though these structures are not clear in Fig. 2c. This difference indicates that the small structures in these PiFM images originated from the wavelength-dependent force, i.e., the photoinduced force. For some regions in Fig. 2b, a strong photoinduced force is observed on both the nanorods and the nanoellipsoids. As can be deduced from the line profile (Fig. 2d), a resolution of ~0.7 nm was achieved in these regions. Note that the line profile of the AFM image ($\Delta z$) does not show extremely small variation, and this indicates that the small area of ~0.7 nm in Fig. 2b did not originate from the artifacts of tip–sample distance control (see the section of Feedback error of the high-resolution imaging in Supplementary Note 3). The reason why the AFM image does not show the small variations is that the decay length of the forces reflecting the AFM image are much longer than that of the PiFM image (see the force curves in Supplementary Fig. 7a and b of the Supplementary Note 2). Furthermore, such tiny variations on the QDs in the PiFM images were observed at different sites and using different tips. In addition to these experimental results, the theoretical calculation also allows the reproduction of such a high resolution. (See the Theoretical analysis of resolution section in the Supplementary Note 12.) Hence, it is verified that PiFM enables substantially high-resolution imaging. To the best of our knowledge, this is the highest spatial resolution achieved thus far for near-field observations in the visible light region. The differences between the images obtained using incident laser lights of 660 and 785 nm can be attributed to the difference in the localised optical response in the QD structure. Therefore, these small areas on the nanorod

might be associated with the defects in the ZAIS QD or crystal material formed during the growth of the nanoellipsoid, which are often observed in the TEM images (see Supplementary Fig. 1b in Supplementary Note 1)[4]. Compared with various techniques of high-resolution scanning nearfield optical microscopy[1–3], PiFM is generally more sensitive[11]. Further, PiFM in UHV using heterodyne-FM is expected to be the most sensitive PiFM technique reported so far[13]. In addition, the sensitivities of local optical responses are enhanced owing to the nanoscale protrusion on the tip, which is generally made of the coating of the tip, as discussed in the Theoretical analysis of resolution section in the Supplementary Note 12. Owing to these facts, it is possible to realise a spatial resolution beyond the single-nanometre-scale.

**3D force field mapping of the photoinduced force.** It is noteworthy that the photoinduced localised potential and force field cannot be visualised via PiFM imaging. Therefore, to visualise these spatial characteristics, we performed 3D photoinduced force mapping. The laser wavelength used to perform the 3D photoinduced force mapping was 660 nm. As shown in the schematics in Fig. 3a, the photoinduced force on the sample surface was mapped by acquiring the force curve ($\Delta f(f_m)X - z$) in feedback mode. Then, the height of the ZAIS QD was adjusted based on the simultaneously acquired AFM image (Fig. 3b inset). Thus, the mapped data were converted into spatial data indicated by the blue rectangular solid in Fig. 3a. Then, we applied a noise reduction technique to the 3D mapping data ($\Delta f(f_m)X(x, y) - z$) based on exploratory factor analysis (see the Exploratory factor analysis section in Supplementary Note 5 for more details)[17–19]. The photoinduced force ($Fz_{pif}$) was obtained by integrating the measured $\Delta f(f_m)X$ as a function of the tip–sample distance ($z$)

with a weight function (see the Frequency shift-to-force conversion in the heterodyne-FM technique section in Supplementary Note 6). Figure 3b illustrates the evaluated force curves ($Fz_{pif} − \Delta z$) of the photoinduced force at the end of the ZAIS QD, in the middle of the QD, and on the gold film as recorded in the mapping. Here, a tip–sample distance of 0 nm ($\Delta z = 0$ nm) is the closest distance between the two, which corresponds to the bottom plane of the blue rectangular solid. At $\Delta z = 0$, $Fz_{pif}$ at the end of the QD is ~7 pN, which is approximately 10 times stronger than that on the thin gold film. This difference is attributed to the self-consistent interactions between the gold gap and ZAIS QD. The attenuation lengths of the force curves are also different; they are ~0.4, ~0.8, and ~1.8 nm at the end of the QD, at the centre of the QD, and on the thin gold film, respectively. These differences in attenuation length are attributed to the self-consistent interactions and lead to sharp force curves[20]. The force curves obtained via theoretical calculations, shown in Fig. 3c, agree well with the experimental ones within the distance range where our calculation method is applicable. Based on our experimental results, an attenuation length of 0.4 nm that is observed from the force curve at the end of the QD is sufficient to obtain the required resolution, as shown in Fig. 2b. Furthermore, it is expected that this attenuation length can enable the visualisation of the interior components of a molecule, which requires a resolution of 1 nm or less. The interaction potential between the tip and sample was obtained as shown in Fig. 3d from the $Fz_{pif}$ mapping, which includes the force curves shown in Fig. 3b. Furthermore, the interaction potential was calculated by integrating $Fz_{pif}$ over $z$[21]; the obtained potential was ~20–40 meV on the nanoellipsoids at $\Delta z = 0$ nm. Here, because the gradient force $\boldsymbol{F}_{grad} \propto \nabla|\boldsymbol{E}|^2$, the potential map reflects the intensity of the electric field (i.e., $U_{grad} \propto |\boldsymbol{E}|^2$).

To obtain variations in the electric field intensity in all directions, we calculate the photoinduced force in the lateral directions. The lateral photoinduced force is obtained by differentiating the interaction potential in the directions parallel to the substrate (i.e., $x$ and $y$). Then, the photoinduced force field vectors in the 3D map can be visualised as Fig. 4[21,22]. Figure 4a, c, e shows the lateral photoinduced force ($Fx, y_{pif}$) for 660 nm wavelength at $\Delta z = 2, 1, 0$ nm, where the colours and direction of the arrows indicate the magnitude and the direction of the force in the $x$ and $y$ direction. Further, Fig. 4b, d, and f visualise the vertical photoinduced force ($Fz_{pif}$) at $\Delta z = 2, 1, 0$ nm. The magnitudes of the forces increase as $\Delta z$ approaches 0 nm. (Similar maps for 520 nm wavelength are indicated in Supplementary Fig. 16 in Supplementary Note 10.) At $\Delta z = 0$ nm, in areas far from the QD, the directions of force fields are random. In contrast, photoinduced force field vectors pointing toward the QD are observed within a range of ~10 nm from the QD. The essential features of vectors near the ZAIS QD in Fig. 4e, and f are quite consistent with the theoretically calculated force vectors in Fig. 4g, h, which further reflect the calculated map of the field-intensity gradient in the absence of the tip (see Supplementary Fig. 20 in Supplementary Note 15 for more details). The results of the theoretical calculations corresponding to these measurements in the Supplementary Information agree well with the essential features of this 3D map. (See Supplementary Fig. 19 in the Supplementary Information.) The agreement between the observed 3D map and the theoretical calculation shows that the present result visualises the successful synthesis of individual QDs to realise the targeted electronic scheme, which demonstrates the effectiveness of our PiFM as a photovisualisation tool.

## Discussion

In summary, we successfully visualised a 3D photoinduced interaction potential distribution and force field vectors; to the best of our knowledge, this is the first time that this has been achieved. The visualisation of photoinduced interaction potential and force field vectors reflecting the photoinduced electric field intensity and its variations are beneficial in quantitatively evaluating the physics of localised optical phenomena related to photocatalyst activities, luminescence. Furthermore, we achieved a resolution of less than 1 nm for our measurements, which is the highest spatial resolution achieved to date for linear optical observations. These high-precision PiFM measurements were achieved using the proposed heterodyne-FM technique under UHV, which enabled measurements with high force sensitivity, resolution, and thermal stability. It is noteworthy that eliminating the photothermal effect induced on the tip by the heterodyne-FM technique is effective even in ambient and liquid conditions. Therefore, the proposed heterodyne-FM technique can be applied in the fields of biology and chemistry. (Application in these fields is expected also considering the present magnitude of the force that can be more enhanced by tightly focused laser spot with high NA lens.) Thus, the achievements in this study indicate the future possibility of observing optical responses inside a single molecule by performing PiFM measurements in a low-temperature environment or using a sharpened tip.

## Methods

**PiFM measurement.** In the heterodyne-FM technique, the photoinduced force has the same phase as the laser modulation. To measure the same phase as the laser modulation, the phase delay due to the electronic circuit used for laser irradiation was compensated by detecting this delay using a photodetector with a rise time of 1 ns. Because of this compensation, a photoinduced force with the same phase as the laser modulation could be detected as a LIX in the heterodyne-FM technique[13]. In this paper, the signal is represented by $\Delta f(f_m)$X. To achieve imaging without the tip degradation during scanning, it is necessary to observe the responses to multiple different wavelengths simultaneously. Then, it is noteworthy that the surface photovoltage on the QD is negligible in this measurement because the 3D size of the QDs is much smaller than the space charge region, and the energies of the incident light, especially 660 and 785 nm, are smaller than the bandgap energies for both the edge and rod of the ZAIS QD[4]. Even if a surface photovoltage existed, the variance in the tip–sample distance for each wavelength per scan arising from the surface photovoltage could be eliminated by carrying out measurements of multiple wavelengths and comparing the obtained images. Here, simultaneous imaging at multiple wavelengths was performed by taking advantage of the heterodyne-FM technique. In the heterodyne-FM technique, a different modulation frequency ($2f_1 + f_{mi}$) can be selected for each incident wavelength ($f_{mi} = f_{m1}, f_{m2}, \cdots, f_{mn}$). Therefore, simultaneous measurements of multiple wavelengths are possible. In these measurements, the modulation frequencies of the laser light ($2f_1 + f_m$) in the heterodyne-FM technique[13] were set to $2f_1 + 230$ Hz ($\lambda_1 = 520$ and 785 nm) and $2f_1 + 325$ Hz ($\lambda_2 = 660$ nm), where $f_1$ is the first resonance frequency of the cantilever. The laser diode was driven such that the laser intensity was $P_{pp} = 15 \pm 15$ mW. For these PiFM measurements, side illumination was used, with an incidence angle of 70°.

**AFM measurement.** The measurements reported in this study were performed using a laboratory-built PiFM apparatus at room temperature in an UHV environment ($<5.0 \times 10^{-9}$ Pa). We used the FM-AFM mode, in which the tip–sample distance is controlled by the shift in the resonance frequency of the cantilever ($\Delta f$)[14]. A gold-coated silicon cantilever (OPUS 240AC-GG, Micromash) with a spring constant of $k \sim 2$ N/m was used in the measurements. The first resonance frequency ($f_1$) was ~44.849 kHz. The Q factor was ~35,400. During the measurements, the driven amplitude ($A_1$) in FM-AFM was maintained at 10 nm. The images shown in Figs. 1 and 2 in the main text were observed in the constant frequency feedback mode ($\Delta f = -20$ Hz). The 3D mapping was performed by acquiring a force curve at each pixel in the image. During mapping, the feedback mode was controlled such that the closest distance ($z_0$) on each force curve would be the tip–sample distance at $\Delta f = -28$ Hz. Then, using the $z_0$ image acquired in the mapping process, the differences in $z_0$ among the force curves were compensated, and the resulting images at the bottom plane are shown in Figs. 3d and 4e, f in the main text.

**Sample preparation.** As reported in the main text, ZAIS QDs on a gold surface were observed as the samples[4]. Each ZAIS QD had a dumbbell shape and a spatially non-uniform electron level structure (Fig. 1b in the main text). The QDs were diluted with toluene to a density of $2.906 \times 10^{-2}$ nM, and then, 20 ml of the QD solution was sprayed onto a thin gold film on mica (AU 15 M, Unisoku).

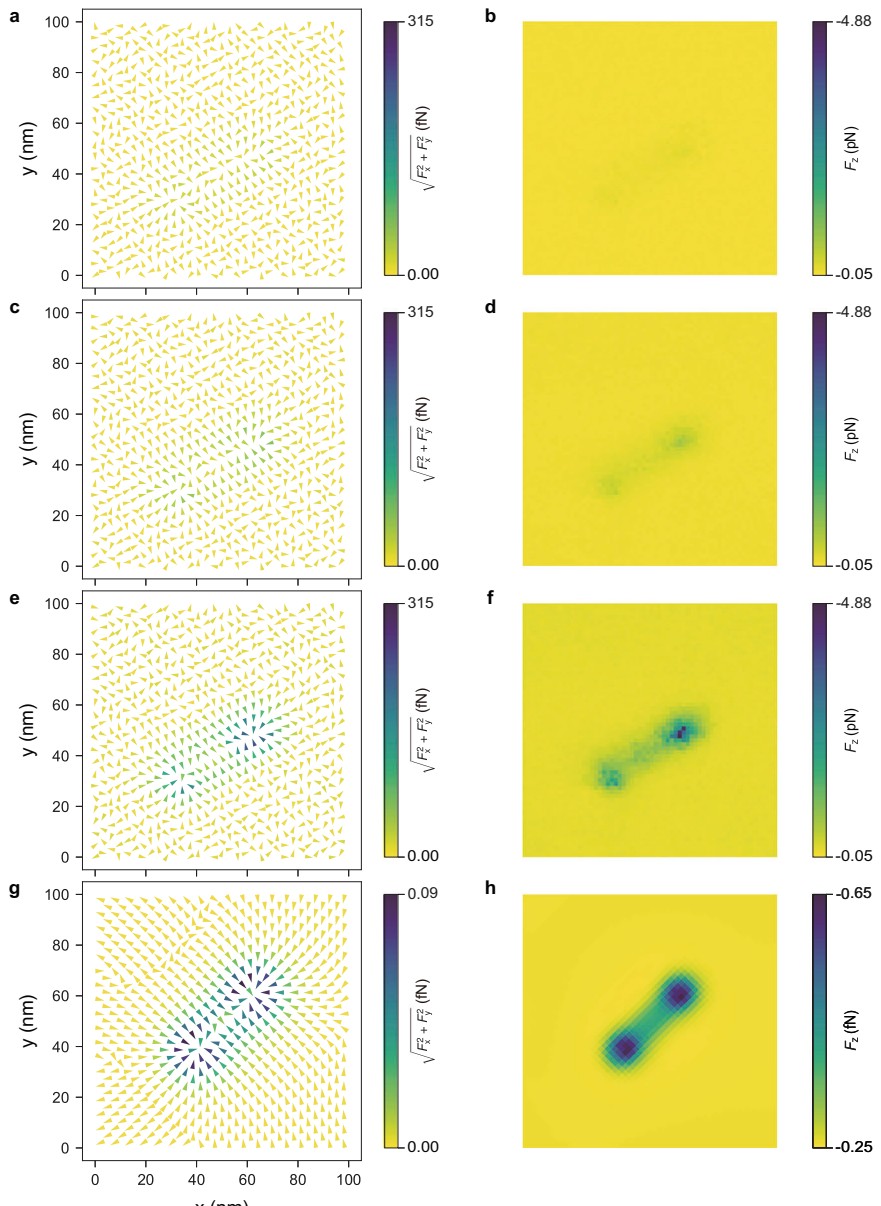

**Fig. 4 Photoinduced force field map using the laser with 660 nm wavelength. a**, **c**, **e** Photoinduced force map directing $x$ and $y$ at $\Delta z = 2.0, 1.0, 0.0$ nm. **b**, **d**, **f** Photoinduced force map directing $z$ at $\Delta z = 2.0, 1.0, 0.0$ nm. **g**, **h** Theoretically calculated photoinduced force field vector mapping ($\Delta z = 1.0$ nm).

## Data availability
The data that support the findings of this study are available from the corresponding authors on reasonable request.

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

## Acknowledgements

This work was supported by JSPS KAKENHI Grant Numbers JP16H06504, JP16H06507, JP16H06327, JP17K19079, and JP16J00304.

## Author contributions

J.Y., Y.N., Y. L., and Y.S. conceived high-precision PiFM measurements. J.Y. developed the heterodyne-FM technique and also performed the experiments and analysed the data. T.K., S.K., and T.T. provided the ZAIS QDs. H.I., N.Y., and H.Y. planned the numerical simulations. H.Y. performed the DDA calculations. J.Y. simulated the thermal expansion that has been expounded in Supplementary information. J.Y., H.Y., H.I., and Y.S. wrote the paper. All authors discussed the results and commented on the manuscript.

## Competing interests

The authors declare no competing interests.
