## [Peer Review File · Nature Communications]

Reviewers' comments:

Reviewer #1 (Remarks to the Author):

The authors report on photoinduced force microscopy (PiFM) of dumbbell-shaped composite quantum dots (QD) with a length around 50nm. The authors aim at determining spatial distributions of photoinduced fields, and they claim to have visualized a photoinduced potential distribution and force field vectors. Such data could benefit the analysis of photocatalysts, for example. Additionally, the authors claim having achieved the highest spatial resolution by PiFM so far, < 1nm.

In theory (and ideally), PiFM is sensitive to optically induced gradient forces. Based on atomic force microscopy the technique requires very careful experimenting to avoid a multitude of possible interferences, susceptibility to photothermal effects being just the most prominent one. Setting up and operating PiFM in such a way that possible interferences are minimized is a task for experts in AFM instrumentation. This audience will be able to fully understand the experimental details provided for reproducing the work.

The authors demonstrate well that the photoinduced signal they measure is not due to photothermal driving of the cantilever, thermal expansion of the sample, or light absorption by the QD. As a result, they focus data analysis and interpretation on the photoinduced gradient force. I wonder if this focus is fully justified, particularly when comparing data derived from actual measurements with simulations.

Surface potential and surface photovoltage:

The dumbbell-shaped composite quantum dots have different chemical composition for the rod and the ellipsoids. I would expect different electronic properties for these parts (surface potential among others). In the presence of possible defects (hypothesized as origin of blobs chosen for resolution claim) or other heterogeneities (expected from TEM image S1, size and possibly composition) one may also expect some variations in the energy diagram.

Surface potential: If not compensated locally, any variation translates into Δf used for topography feedback, hence height errors arise that will affect the magnitude of the gradient force detected. This would also affect the ratios in Fig. 1g,k.

Surface photovoltage: Appears to be uncompensated in the setup described. Is it justified to neglect surface photovoltage? Would it not affect the gradient force detected? At least the green wavelength is above the bandgap of the ellipsoids (according to Fig. 1b), and variations in actual energy levels and gaps may be expected.

Please address these issues.

Resolution:

Fig. 2b, to a lesser extent also Fig. 2c, exhibit small blobs that correlate over a few scan lines and may hint at a resolution on the order of 1nm. However, is it not a bit odd that these features appear on topographic edges/slopes, regions that notoriously create 'features' in scanning probe microscopy? Fiddling a bit with brightness and contrast of the corresponding topography scan (unfortunately at the end of the color scale), some of the features on the ellipsoid and rod also become discernible.

As an old-school microscopist I find resolution claims that can be verified by independent means much more convincing. Could it be that the actual shape of the tip supports local field enhancement at 660 nm but not as much at 785 nm? Would it count as spatial resolution if tip features cause the signal, i.e. tip features are imaged? What is the reason for the observed wavelength dependence? According to Fig. 1, neither by the measurements nor by the results of the discrete dipole approximation such a difference is immediately evident.

Please strengthen the claim for actual resolution.

Three-dimensional field and potential distribution:

Side remark: 'stereoscopic distributions' sounds a bit odd; 'spatial' or 'three-dimensional' distributions may be more appropriate.

Clearly, three-dimensional mapping of the photoinduced gradient force is the type of measurement required. As for all 3D force mapping, assigning the correct ground level (e.g. minimal separation between tip and

sample) is critical. Hence it is essential to properly handle contact potential differences and photoinduced voltages. I must admit I was a bit disappointed that instead of a visualization of the 3D distribution of the photoinduced force field the authors only present three force curves acquired at relevant locations and a vector mapping on the ground plane. This presentation falls short of the 'wish list' and justifications for the need of 3D field distributions given at the beginning of the paper.

Please improve. It would be really interesting to see the evolution of the field above the ground plane. I am much less worried about the huge mismatch between measured and calculated force values. At this stage of PiFM (and scanning probe microscopy in general) there are still plenty of factors related to the probe that, unfortunately, can be accounted for by plausible arguments only. Nevertheless, a qualitative picture alone would already be very helpful.

In short, an interesting contribution, particularly for the scanning probe microscopy community, with room for further strengthening.

Reviewer #2 (Remarks to the Author):

In this paper the authors seek to visualize a photoinduced electric field distribution using composite quantum dots that have tailored electronic and optical structures--such as photocatalysts--at the single-nanometre scale using photoinduced force microscopy (PiFM). PiFM was used to measure the photoinduced interactiong potential gradient force which should be proportional to the gradient of the electric field intensity in the vicinity of the structure being imaged.

One significant limitations of this approach is the use of a high vacuum which renders the method inapplicable for any biological problems. The authors do not provide any discussion about what spatial resolutions are possible with competing techniques. Some discussion of spatial resolution can be found in references 1-10. However, since the point of the measurements presented here is spatial resolution, i.e. "single nanometer scale" from their manuscript title, the authors must explicitly compare to existing techniques. Without such comparison the reader cannot judge whether the method described in the manuscript is an advance over existing approaches. The concept of resolution of an imaging system has specific formal interpretations in the literature dating back to Rayleigh's work. The authors in this manuscript don't provide any quantitative discussion of resolution but simply refer to the width of a quantum dot. If the quantum dot is small enough, this could provide the point response function for the system. Since it's not quite small enough for that, the image of the QD will be the convolution of the object and the point response function. It is possible to deduce something about spatial resolution but some more analysis or at least a longer discussion is required.

Reference 21 was not cited in the manuscript anywhere.

typo: The agreement between the observed 3D map and the theoretical calculation ***shows*** that the present result visualizes the successful synthesis of individual QDs to realize targeted electronic scheme

Response to Reviewer #1

We are grateful to the reviewer for very carefully reviewing our manuscript and for valuing our results. We are greatly encouraged by the reviewer's positive evaluation of our work. We have read the reviewer's comments and questions very carefully and have found them to be crucial for improving our manuscript. Further, it appears that some of the concerns of the reviewer have arisen owing to our inadequate presentation of our study in the previous manuscript. Therefore, we have revised our manuscript in accordance with the reviewer's criticism. We sincerely hope that the revised manuscript will be suitable for publication in *Nature Communications*. Our detailed point-by-point responses to the reviewer's comments are as follows.

Reviewer's Comment 1:

The authors demonstrate well that the photoinduced signal they measure is not due to photothermal driving of the cantilever, thermal expansion of the sample, or light absorption by the QD.

Our response:

We are delighted that the reviewer appreciates our demonstration, which successfully avoids the photothermal effect on the measured signal. This technique is the focal point of our demonstration for obtaining high resolution 3D force mapping.

Reviewer's Comment 2:

As a result, they focus data analysis and interpretation on the photoinduced gradient force. I wonder if this focus is fully justified, particularly when comparing data derived from actual measurements with simulations.

Surface potential and surface photovoltage:

The dumbbell-shaped composite quantum dots have different chemical composition for the rod and the ellipsoids. I would expect different electronic properties for these parts (surface potential among others). In the presence of possible defects (hypothesized as origin of blobs chosen for resolution claim) or other heterogeneities (expected from TEM image S1, size and possibly composition) one may also expect some variations in the energy diagram.

Surface potential: If not compensated locally, any variation translates into Δf used for topography feedback, hence height errors arise that will affect the magnitude of the gradient force detected. This would also affect the ratios in Fig. 1g, k.

Surface photovoltage: Appears to be uncompensated in the setup described. Is it justified to neglect surface photovoltage? Would it not affect the gradient force detected? At least the green wavelength is above the bandgap of the ellipsoids (according to Fig. 1b), and variations in actual energy levels and gaps may be expected.

Please address these issues.

Our response:

We thank the reviewer for this crucial comment. As the reviewer correctly points out, the surface potential and surface photovoltage can affect the feedback control, namely, a small error might appear with regard to the height adjustment of the probe because of a small shift in the frequency arising from the shifting of the electrostatic-force gradient. However, the error arising from the surface potential hardly affects the observed PiFM image because the interaction force in the feedback control is the van der Waals force between the sample and probe tip. The electrostatic force due to the surface potential is quite small as compared with the van der Waals force because of the almost zero contact potential difference between the gold tip and gold substrate, and hence, it is reasonable to consider the height error to be much less than the height range where the change of the field gradient force is visibly detectable. This point is very clear from the force curves at the rod and the ellipsoid in Fig. S7 in the Supplementary Information. The two curves totally coincide with each other at the regions where the images were taken. Specifically, the influence of the surface potential on the PiFM image is not problematic. Therefore, the influence of the surface potential on the ratios (between different parts of the sample) seen in Fig. 1 g, k does not change the essence of our observations. Although it is not easy to estimate a definite value of the height error in the feedback control in the present technique, our discussion in the manuscript is validated by the observation of the force curves in Fig. S7 and by good agreements between the observed results and theoretical calculation.

In addition to the above consideration, it is important that the heterodyne-FM technique, which is proposed in this paper, enables us to measure using multiple wavelengths in one scanning. Therefore, the images that we compared were measured

completely at the same time, and hence, the tip-sample distance variation is entirely the same and the feedback control should not affect the comparison between the images for the different wavelengths even if surface potential and surface photovoltage on the QD exist. This is a strongpoint of our measurement technique.

Regarding the surface photovoltage, as the reviewer suggested, when the light enters the semiconductor surface, the generated carriers relax the band bending at the surface, which generally induces the surface photovoltage. This effect might change the contact potential difference, influencing the electrostatic force between tip and sample. In the case of the bulk, the surface photovoltage variation in bulk semiconductors due to the separation charge region is ~ 100 nm. However, this length is substantially longer than the QD size. Therefore, surface photovoltage should not generate on the surface of the QDs used in the present study.

Owing to the aforementioned reasons, we conclude that surface potential and surface photovoltage do not affect our observations and the ensuing discussion. However, the points that the reviewer raised are quite important and essential for readers to understand the significance of our results, and we became aware that more detailed descriptions on these points are desirable. Therefore, we have added the sentences that explain the above-mentioned points in p.2, from line 60 in the right column to line 76 in the right column, and in the Methods section (p.4, left column, lines 26 to 34).

Reviewer' s comment 3

Resolution:

Fig. 2b, to a lesser extent also Fig. 2c, exhibit small blobs that correlate over a few scan lines and may hint at a resolution on the order of 1nm. However, is it not a bit odd that these features appear on topographic edges/slopes, regions that notoriously create 'features' in scanning probe microscopy? Fiddling a bit with brightness and contrast of the corresponding topography scan (unfortunately at the end of the color scale), some of the features on the ellipsoid and rod also become discernible. As an old-school microscopist I find resolution claims that can be verified by independent means much more convincing. Could it be that the actual shape of the tip supports local field enhancement at 660 nm but not as much at 785 nm? Would it count as spatial resolution if tip features cause the signal, i.e. tip features are imaged? What is the reason for the observed wavelength dependence? According to Fig. 1, neither by the measurements nor by the results of the discrete dipole

approximation such a difference is immediately evident. Please strengthen the claim for actual resolution.

Our response:

We thank the reviewer for this crucial comment. We consider that the reviewer’s concern regarding our discussion of the high-resolution image with regard to Fig. 2 mainly includes following four points: (1) One is about the crosstalk between the PiFM and AFM images. (2) The second point says that resolution claims that can be verified by independent means are much more convincing. (3) The third is the influence of tip features on the PiFM images. (4) The fourth point is about the reason behind the frequency dependences of images. We address these points one by one, as follows.

(1) As the reviewer had mentioned, it is possible that the photoinduced force signal affects AFM images. This is because our heterodyne-FM technique requires the laser power modulation to be $P_0(1 + \cos\{(2\omega + \omega_m)t\})$. The DC component P_0 constitutes the DC component of the photoinduced force. Therefore, if a strong photoinduced force is exerted on the tip, the photoinduced force will appear in feedback control. Although it is difficult to remove this effect completely using the present technologies, this issue is not critical with regard to our measurement. To confirm that the “small blobs” in the PiFM images did not originate from the forces appearing in the AFM images, we evaluated the photoinduced force at the sites where the “small blobs” do not appear in the topography image (Figure A(d)). This indicated that the photoinduced force in the line profile (Fig. A(d)) did not originate from the other forces observed in the AFM image. This has been discussed in the main text, and we have replaced Fig. 2 with the figure presented below (Figure A). In addition, the contrast of the “small blobs” relative to the other sites, i.e., rod, edges, and gold substrate, is more obscure at 785 nm than at 680 nm in Fig. A(b). These wavelength-dependent variations indicate that the “small blobs” originated from a wavelength-dependent optical interaction. Thus, we confirmed that no problem arises owing to the crosstalk between the PiFM and AFM images in our measurement.

Figure A

(2) Regarding the verification of the resolution by independent means, we have carried out a theoretical examination. By using the extended version of discrete dipole approximation (extended DDA) developed by us (refer to the theoretical part of “Takase et al, Nature Photonics 7, 550-554 (2013)”), we have calculated the gradient of the photoinduced force exerted on the tip (which is approximately proportional to the amount measured via PiFM). In this method, we can model the sample structure and evaluate the change in the field intensity in an angstrom scale for a particular portion of the simulation volume using the multiple-cell-size method. (In the main text, we did not discuss this method as it may lead to a heavy cumulative computational load.) We assumed that protrusions with a diameter of approximately 1 nm exist on the sample surface and calculated the gradient of the force when the tip-sample distance was 0.4-0.5 nm. For this distance, we can use the classical model for the metallic tip safely because there is no overlap of wavefunctions between the tip and sample (See, R. Esteban, et al., Faraday Discuss., 2015, 178, 151.) We examined the resolution under two scenarios with regard to the tip. One is the case where there are no protrusions on the tip, and the other is the case where there are atomic scale protrusions on the tip. (See Fig. B(a,b) shown below for the models of the tip and sample.) It is well-known that at the tip of the gold-coated probe, there are several atomic scale protrusions; hence, the assumption that the latter case is accurate is reasonable. It is not essential to consider the tip-by-tip differences in the shapes of the protrusions, as can be understood from the following discussion: Please note the following two important points elucidated by the numerical calculations. [1] Even in the case where there are no protrusions on the tip, some structures appear in the line profile of the gradient of the force. (See Fig. B(f). Here, we limit the window of the horizontal axis in order to eliminate the calculation artefacts arising from the step structures on the probe model.) Notably, in the presence of atomic scale protrusions on the tip, very clear structures appear in the line profile, which is very much in line with our observed results (See Fig. B(e)). This is because the gradient of the force is substantially sensitive to the local change of the field intensity. Assuming that the existence of atomic scale protrusions on the tip is reasonable, the actual situation is more similar to Fig. B (a) than to Fig. B (b). This adequately explains why we can obtain a sufficient resolution to see individual protrusions on the sample. [2] Because of the finite curvature of the probe tip, it is reasonable to consider that one protrusion projects outward the farthest and that the distances between the sample and the other protrusions are therefore not the same as that between the farthest protrusion and the sample. Considering such a situation,

as seen in Fig. B (a), we assume that there are many other protrusions whose heights are very similar to that of the central protrusion (the difference is only 0.2 nm in height, as indicated by Fig. B (a)). However, there is no occurrence of ghosts, and the central protrusion alone is sensitive to protrusions on the sample, as shown in Fig. B (e).

Considering [1] and [2], we can conclude that our theoretical examination successfully verifies the resolution of our method. We hope that the above examinations convince the reviewer and readers that a resolution of ~ 0.7 nm was achieved in our experiments.

Figure B

(3) It is not necessary to be concerned about the possibility that tip features may appear in the image because ghost features do not appear in the PiFM image unless the tip contacts the sample. As the reviewer pointed out, ghost features can affect the PiFM image if the tip contacts the samples. However, in our case, the tip did not contact the sample. In the present measurements, we carefully controlled the tip such that it did not contact the samples using FM-AFM, which is a non-contact mode of performing AFM measurements. Further, these high-resolution images were obtained even when we used other tips and QDs as shown in Fig. C below.

Figure C

Here, Fig. C (a) and Fig. C (c) are AFM images of QDs. Fig. C (b) and Fig. C (d) are PiFM images obtained with a 660 nm laser. This constitutes strong evidence that the high-resolution images are not due to a specific tip feature. The contrast differences between the images obtained with 660 nm and 785 nm lasers are also observed, as indicated below (Figure D). Here, Fig. D (b) and Fig. D (c) are the images obtained using 660 and 785 nm lasers, respectively.

Figure D

From these facts, we conclude that the measured PiFM images do not reflect the tip features.

(4) We explain the frequency dependences of the images as follows: The PiFM image obtained with a 785 nm laser shows weaker contrast of the PiFM signal on the small structures than that obtained using a 660 nm laser. The optical gradient force between the tip and sample is approximately $F \propto \text{Re}[\alpha_{\text{tip}}] \text{Re}[\alpha_{\text{sample}}]$. When we use a laser, the

gradient force is influenced by the term $\text{Re}[\alpha_{\text{tip}}]$. However, this term is constant throughout the scan area, and the frequency dispersion is small in this frequency region. Therefore, the contrast of the scan image comes only from $\text{Re}[\alpha_{\text{sample}}]$, that is, the QD optical feature at the specific site. Therefore, the differences between the PiFM images obtained using 660 nm and 785 nm lasers originated from the local optical response. In fact, we have provided the corresponding images in the response for (3).

We hope that concerns (1)-(4) expressed by the reviewer have been fully addressed through the above discussion. In addition, we consider that the points raised by the reviewer and the related discussion will be very helpful and important for the readers. Therefore, we have added sentences discussing points (1), (3), and (4) in the manuscript (p. 2, right column, lines 91 to 94 and 98 to 109). In addition, we have added the figures shown above (Figure C) in the Supplementary Information. With regard to point (2), we have added sentences discussing it as well in the Supplementary Information.

Reviewer' s comment 4

Three-dimensional field and potential distribution:

Side remark: 'stereoscopic distributions' sounds a bit odd; 'spatial' or 'three-dimensional' distributions may be more appropriate. Clearly, three-dimensional mapping of the photoinduced gradient force is the type of measurement required. As for all 3D force mapping, assigning the correct ground level (e. g. minimal separation between tip and sample) is critical. Hence it is essential to properly handle contact potential differences and photoinduced voltages. I must admit I was a bit disappointed that instead of a visualization of the 3D distribution of the photoinduced force field the authors only present three force curves acquired at relevant locations and a vector mapping on the ground plane. This presentation falls short of the 'wish list' and justifications for the need of 3D field distributions given at the beginning of the paper. Please improve. It would be really interesting to see the evolution of the field above the ground plane. I am much less worried about the huge mismatch between measured and calculated force values. At this stage of PiFM (and scanning probe microscopy in general) there are still plenty of factors related to the probe that, unfortunately, can be accounted for by plausible arguments only. Nevertheless, a qualitative picture alone would already be very helpful.

Our response:

We thank the reviewer for these especially important comments and suggestions.

According to the reviewer's advice, we have revised the term 'stereoscopic distribution' to 'three-dimensional distribution'.

We particularly appreciate the following comment made by the reviewer: 'As for all 3D force mapping, assigning the correct ground level (e.g. minimal separation between tip and sample) is critical. Hence it is essential to properly handle contact potential differences and photoinduced voltages'. We consider that we should have explained this point in detail in the main text. The contact potential differences do not constitute a problem as simultaneous measurements are performed via AFM and PiFM, and the photoinduced voltages are negligible because of the large volume of the charge separation region (please see our response to comment 2 for more detail). In response to comment 2, we have added sentences that explain the above-mentioned points in the Methods section in the main text (p4, left column, lines 26 to 34).

In addition, we believe that the comment "This presentation falls short of the 'wish list' and justifications for the need of 3D field distributions given at the beginning of the paper" is very helpful for improving our manuscript. We have added and revised some sentences in this regard in p.1, left column, lines 17 to 19. Furthermore, the comment 'It would be really interesting to see the evolution of the field above the ground plane' has strongly motivated us to present a more elaborate 3D map, which is quite helpful for improving our manuscript. We have replaced Fig. 4 with the figure shown below (Figure E). Moreover, we have added the corresponding figure to Fig. Ea regarding the theoretical calculation in the Supplementary Information in addition to adding a few sentences in p. 3, right column, lines 73 to 83.

Figure E

Further, we thank the reviewer for the comment on the substantial mismatch between the experimental data and the theoretical calculations of the strength of the photoinduced force in the main text. The experimental result of the photoinduced force is 3500 times larger in the horizontal direction and 9600 times larger in the vertical direction compared to the theoretically calculated photoinduced force. Although we discussed this problem in the “Magnitude of photoinduced force” section in the previous Supplementary information, we have found the argument there to be insufficient owing to the reviewer’s comment and have provided a better discussion.

In the present study, the fitting of the absolute value is not crucial because there are several factors of the tip that do not affect the essence of the present discussion; in addition, the light intensity near the sample is uncertain. (The force is simply proportional to the light intensity.) Nevertheless, we can provide a plausible discussion as follows: In the main text, we showed the results of the photoinduced force calculated by assuming that the tip had a diameter of 19 nm because of the calculation cost. In contrast, in the aforementioned supplementary section, we had presented the result for a tip with a diameter of 60 nm, which is similar to the size of the tip used in the experiment, where the photoinduced force becomes approximately 60 times larger. In this calculation, the minimum tip-sample distance is 1 nm because

of the cell size in the discrete dipole approximation in the present simulation. In the experimental data, the photoinduced force at 0 nm is 10 times larger than that at 1 nm. When these factors are multiplied, the force increases by approximately 600 times. Another factor is the shape. In our calculation, the tip was assumed to be spherical. If we consider the length of the tip, the force becomes about three times larger according to our calculation. (In the previous supplementary section, we had stated this to be ‘ten times larger’ according to a simple estimation. We have corrected this in the revised version according to a more elaborate calculation. If we also consider the uncertainty of light intensity, the present discrepancy in the induced force between the calculated and observed results ceases to be a problem (though the laser intensity of 10 kW/cm^2 that we used in the calculation is reasonable). Furthermore, if we re-evaluate all the images using the extended DDA method explained above, we can expect to obtain different force values. However, we consider that such a trial is not very fruitful under a situation where the incident light intensity is not exactly known.

In the revised version of the Supplementary Information, we have improved our explanation by using the results obtained via more elaborate calculations.

Response to Reviewer #2:

We would like to thank the reviewer for very carefully reviewing our manuscript and for the invaluable comments and criticism regarding our work. We believe that the suggestion for discussing the ‘spatial resolution’ has proved to be particularly important for improving our manuscript. This suggestion has highly motivated us to examine and discuss this issue in substantial detail. Consequently, we strongly believe that we have successfully achieved the highest resolution reported thus far,

as confirmed via both experimental and theoretical methods elucidated in the subsequent paragraphs. This was indeed possible owing to the crucial comments made by the reviewer. We greatly appreciate his/her **careful** review of our manuscript.

According to the reviewer's comments and criticism, we have revised our manuscript very carefully, and our point-by-point responses to the reviewer's comments are as follows. We sincerely hope that this response and the revised manuscript will adequately address all the concerns raised by the reviewer and that the revised is suitable for publication in *Nature Communications*.

Reviewer' s comment 1

One significant limitations of this approach is the use of a high vacuum which renders the method inapplicable for any biological problems.

Our response:

We thank the reviewer for this comment. As the reviewer has pointed out, presently, the experiment is performed in vacuum. Further, as explained in the main text, the proposed method is a powerful technique for observing physical phenomena with an ultimate (atomic scale) resolution because of the force sensitivity of the cantilever as well as the thermal stability of the PiFM measurements. However, we would like to emphasize that the advantages of this method, which is realized using the heterodyne-FM technique that enables us to isolate the photothermal expansion from the PiFM signal, are fully effective in ambient or liquid conditions. Additionally, the photoinduced force in liquids (which is several tens of times higher than that in vacuum) is definitely detectable in our method. Hence, our technique can be extended and applied in biology with an appropriate resolution. Such applications are the subject of our future research. Owing to this comment, we have become aware that providing some comments on the potential of our method in this regard will benefit our readers in understanding the full potential of our proposed scheme; thus, we have added a few relevant sentences in p.4, left column, line 3 to p.4, left column, line 8.

Reviewer' s comment 2

The authors do not provide any discussion about what spatial resolutions are possible

with competing techniques. Some discussion of spatial resolution can be found in references 1-10. However, since the point of the measurements presented here is spatial resolution, i. e. “single nanometer scale” from their manuscript title, the authors must explicitly compare to existing techniques. Without such comparison the reader cannot judge whether the method described in the manuscript is an advance over existing approaches.

Our response:

We thank the reviewer for this helpful comment. It is true that the comparison between existing methods and our method in terms of the resolution is important for readers to understand the substantial advancement that is represented by our method. In references 1-3, various techniques of high-resolution scanning nearfield optical microscopy (SNOM) are discussed. These methods detect photo signals. Compared with them, PiFM is generally more sensitive because it detects induced forces, which do not suffer from any type of attenuation like photo-signals do. This advantage has been well-discussed in research articles such as ‘Applied Physics Letters 97, 073121 (2010)’. Further, our experiment is performed in UHV, which greatly enhances the sensitivity, as discussed in our previous paper (Physical Review Applied 9, 024031 (2018)). The demonstration described in the present paper was the first trial to perform PiFM in UHV; further, the PiFM performed in this study by using heterodyne-FM to eliminate the photothermal effect is expected to be the most sensitive PiFM reported thus far. Owing to these reasons, we were able to achieve the highest resolution reported to date. According to the suggestion of the reviewer, we have added a few sentences to explain the above-mentioned comparison in the main text (from p.3, left column, line 2 to p.3, left column, line 10).

Reviewer’ s comment 3

The concept of resolution of an imaging system has specific formal interpretations in the literature dating back to Rayleigh’ s work. The authors in this manuscript don’ t provide any quantitative discussion of resolution but simply refer to the width of a quantum dot. If the quantum dot is small enough, this could provide the point response function for the system. Since it’ s not quite small enough for that, the image of the QD will be the convolution of the object and the point response function. It is possible to deduce something about spatial resolution but some more analysis or at least a longer discussion is required.

Our response:

We greatly appreciate this comment because the point raised by the reviewer is a central issue that should be discussed. This comment is definitely helpful for improving our manuscript. As pointed out by the reviewer, the image of the QD is obtained as the convolution of the object and the point response function. We have information regarding the optical susceptibility of quantum dots, deduced from the experimentally observed absorption spectra. Thus, we can theoretically evaluate this image via a simulation that directly includes the effect of the point response function and the explicit spatial geometry of objects, which provides the most direct quantitative evaluation of spatial resolution.

To this end, we used the extended version of discrete dipole approximation (extended DDA) developed by us (see, the theoretical part of ‘Takase et al, Nature Photonics 7, 550-554 (2013)’). We have calculated the gradient of the photoinduced force exerted on the tip (which is approximately proportional to the amount measured via PiFM). In this method, we can model the sample structure and evaluate the change in the field intensity in an angstrom scale for a particular portion of the simulation volume using the multiple-cell-size method. We assumed that protrusions with a diameter of approximately 1 nm exist on the sample surface and calculated the gradient of the force when the tip-sample distance was 0.4-0.5 nm. For this distance, we can use the classical model for the metallic tip safely because there is no overlap of wavefunctions between the tip and sample (See R. Esteban, et al., Faraday Discuss., 2015, 178, 151.) We examined the resolution under two scenarios with regard to the tip. One is the case where there are no protrusions on the tip, and the other is the case where there are atomic scale protrusions on the tip. (See Fig. A (a,b) shown below for the models of the tip and sample.) It is well-known that at the tip of the gold-coated probe, there are several atomic scale protrusions; hence, the assumption that the latter case is accurate is reasonable. It is not essential to consider the tip-by-tip differences in the shapes of the protrusions, as can be understood from the following discussion: Please note the following two important points elucidated by the numerical calculations. [1] Even in the case where there are no protrusions on the tip, some structures appear in the line profile of the gradient of the force. (See Fig. A(f). Here, we limit the window of the horizontal axis in order to eliminate the calculation artefacts arising from the step structures on the probe model.) Notably, in the presence of atomic scale protrusions on the tip, very clear structures appear in the line profile, which is very much in line with our observed results (See

Fig. A(e)). This is because the gradient of the force is substantially sensitive to the local change of the field intensity. Assuming that the existence of atomic scale protrusions on the tip is reasonable, the actual situation is more similar to Fig. A (a) than to Fig. A (b). This adequately explains why we can obtain a sufficient resolution to see individual protrusions on the sample. [2] Because of the finite curvature of the probe tip, it is reasonable to consider that one protrusion projects outward the farthest and that the distances between the sample and the other protrusions are therefore not the same as that between the farthest protrusion and the sample. Considering such a situation, as seen in Fig. A (a), we assume that there are many other protrusions whose heights are very similar to that of the central protrusion (the difference is only 0.2 nm in height, as indicated by Fig. A (a)). However, there is no occurrence of ghosts, and the central protrusion alone is sensitive to protrusions on the sample, as shown in Fig. A (e).

Figure A

Considering [1] and [2], we can conclude that our theoretical examination successfully verifies the resolution of our method. We hope that the examinations elucidated above convince the reviewer and readers that a resolution of ~ 0.7 nm was achieved in our experiments. We have added the relevant discussion on this point in the main text (p.2, right column, lines 98 to 109) and also elaborated on this in the Supplementary Information.

Reviewer' s comment 4

Reference 21 was not cited in the manuscript anywhere.

Our response:

We thank the reviewer for pointing out this error. We have corrected it by citing this reference (21) at p.3, right column, line 73.

Reviewer' s comment 5

typo: The agreement between the observed 3D map and the theoretical calculation *****shows***** that the present result visualizes the successful synthesis of individual QDs to realize targeted electronic scheme

Our response:

Thank you for pointing out this typographical error. We have corrected it.

REVIEWER COMMENTS

Reviewer #1 (Remarks to the Author):

Thank you for carefully addressing the concerns raised and providing further details in the manuscript.

Reviewer #4 (Remarks to the Author):

This is an interesting paper that has substantial impact potential if the issues highlighted below are addressed. It also has already gone through 2 rounds of refereeing, most issues have been addressed.

In the flowing, figures refer to the figure labels provided in response to the other referees, except where indicated. My major issues that should be addressed are **highlighted below**.

1. What are the noteworthy results?

Visualization in 3D of the photoinduced electric field with nm scale resolution is the stated aim of this paper. This is an important result, with wide potential impact, as summarized well in the manuscript.

Fig 4 in the manuscript shows the main result. This figure is very hard to read – what are the gray scales and color scales in 4b? Contrast of non-blue arrows in 4a very weak. Given that the blue short arrows in 4a are essentially background – why plot them? If this paper is to be published the major aim of it (3D mapping of optically induced fields) needs to be presented much better. I would have loved to see a 3D field as a result of different wavelengths of illumination – maybe even a difference map?

I think the stated aim of the paper has been demonstrated – but it needs to be presented much better. The measurements also open many questions, which should either be discussed or mentioned for future work.

2. Will the work be of significance to the field and related fields? How does it compare to the established literature? If the work is not original, please provide relevant references.

Nanometer scale imaging of light induced electrical fields is important to understand many phenomena – catalysis, biological processes etc. As such, a demonstration in UHV is a good first step – I would like to see more of a discussion of how the technique presented will be transferrable to liquids or gas, given the well-known problems of PiFM contrast.

What I find worrisome in this context is the fact that the measured signal is 100,000 times smaller than the expected PiF signal. Plausibility arguments are made for why one is so far off - but this should be more clearly stated in the paper. Some of the assumptions could be tested by careful experiments (and should be done down the road). This small signal does not bode well for transferring this technique to liquids or gas, where AFM sensitivity is drastically reduced. It also leads to a large sensitivity to other sources of PiF ‘signals’ not directly related to electrical fields. **This discrepancy should be stated more clearly as a challenge.**

If one wanted to be really critical, one could state that contrast on the 1nm level is observed due to light shining on a sample, the origin of which is unclear, but possibly due to PiF. This

however would be more of a criticism of the current understanding of contrast in the PiFM field as a whole than the original and interesting results presented in this paper.

3. Does the work support the conclusions and claims, or is additional evidence needed?

This is where I have some major questions, already raised by previous referees, and in my opinion not yet answered adequately in the manuscript.

Origin of nanoscale contrast:

I am not convinced that light induced contact potential differences do not play a role – why is NO surface photovoltage (SPV) observed is very puzzling. This group has great experience in performing KPFM – **I would only accept to publish this paper if I saw KPFM data on nanorods with/without light.** Why not just show data and settle this discussion once and for all? Also, CPD data would allow a better determination of the minimal separation between tip and sample – crucial for the generation of 3D plots of the E-field. The volume of charge separation is not huge – in particular if one argues 1nm spatial resolution...

Potentially all the contrast comes from surface potential differences. Variations in SPV could explain the observed highly localized spots – traps being photo deactivated? The interpretation of the data would then change dramatically. In particular, comment 2 of referee 1: *'Surface potential: If not compensated locally, any variation translates into Δf used for topography feedback, hence height errors arise that will affect the magnitude of the gradient force detected. This would also affect the ratios in Fig. 1g,k.'* is not answered satisfactory.

The author response: *'Therefore, the images that we compared were measured completely at the same time, and hence, the tip-sample distance variation is entirely the same and the feedback control should not affect the comparison between the images for the different wavelengths even if surface potential and surface photovoltage on the QD exist. This is a strong point of our measurement technique.'* This is only true if Au and the nanorod have the same CPV – highly unlikely! If different wavelengths activate traps or lead to spatially and temporally varying SPV on the nanorod then the deduced electric field is a convolution of several physical phenomena.

Figure S7 is not sufficient to demonstrate the absence of KPFM or SPV contrast – I want to see KPFM parabolas with and without light over the different regions as well as KPFM with/without light simultaneously to the data presented. I would in fact expect a difference in SPV!

In their answer, the authors state *'In the case of the bulk, the surface photovoltage variation in bulk semiconductors due to the separation charge region is ~ 100 nm. However, this length is substantially longer than the QD size. Therefore, surface photovoltage should not generate on the surface of the QDs used in the present study.'* This is correct for equilibrium situations. However, empirically, many groups have observed differences in KPFM with/without light in monolayers of organic molecules (where the argument above would apply; hence the request for empirical SPV data.

In passing, I observe that S7 is used to argue (correctly, in my opinion) that *'the large difference in n indicates that the detected PiFM signal does not reflect the thermal expansion.'* But - what proof is there that vdW dominates? How big is the change of CPD between Au and NP?

4. Are there any flaws in the data analysis, interpretation and conclusions? - Do these prohibit publication or require revision?

I agree that one clearly observes some spatial variation of the signal on the order of 1nm. I think the authors should be more careful in arguing the origin of this resolution. Stating the open questions will stimulate more research.

Three different lines of argument for the ~1nm resolution are considered by the authors in response to comment 3 of referee 1 on the resolution:

1. Cross talk:

Referee 1 states *'Fiddling a bit with brightness and contrast of the corresponding topography scan (unfortunately at the end of the color scale), some of the features on the ellipsoid and rod also become discernible.'* Figure A in the response seems convincing, but how can one exclude cross-talk between channels, e.g. due to feedback imperfections? Only the feedback signal (i.e. the black line in A) is shown. How big is the feedback error signal?

2. independent verification of resolution:

Here, essentially nm scale roughness on the tip is used as a model to explain the source of the high spatial resolution. Although I sympathise with the model, it has several potential flaws:

- a) 0.5 nm is a close distance, typically used for tunneling, which is due to wave function overlap. So a classical model (as argued) does not apply.
- b) Why should a gold tip support 1nm rough features (or *'atomic scale protrusions'* ...)? Gold has a high diffusion constant – such a tip would not be stable, in particular when illuminated by light. I disagree with the authors statement *'As is well known, the tip of a gold-coated probe has many atomic-scale protrusions.'* – all published TEM images of Au tips I am aware of show that they are smooth on a nm scale. References for the statement made? In the supplemental section *'The assumption of the existence of atomic scale protrusions on the tip is reasonable, and we consider that the actual situation is closer to that depicted by Fig. S12 than that depicted by Fig. S12b;'* In fact I would argue that the tip in Fig S12b is more realistic. Give me references for why is this a *'reasonable'* argument! (The answer cannot be that the modelling supports the experiment, as this is a circular argument not supported by independent evidence).

It is stated that the contrast is dominated by the front tip protrusion.

In contrast to the model, the sample does not seem to show topographical variation.

What then is the physical origin of the spatial variation of properties leading to the stated ~ 1 nm spatial resolution? This is also where local KPFM contrast (=material contrast) would provide corroborating evidence.

In the Suppl. Section the authors state '*Considering such a situation, as seen in Fig. S12a, we assume that many other protrusions exist whose heights are very similar to that of the central protrusion (the difference is only 0.2 nm in height). However, no ghost appears, and the central protrusion alone is sensitive to protrusions on the sample.*'

I think this is potentially a good test. If the contrast essentially disappears for a 0.2 nm distance change - how does the contrast in S12a change for a change of 0.2 nm in distance?

- How does it change if the tip-sample distance is changed by 0.2 nm in the model?
- Does the contrast experimentally disappear for a change of 0.2 nm in tip-sample separation?
- What if the feedback error is 0.2 nm?

The authors argue with a variation of $\text{Re}[\alpha_{\text{sample}}]$, and assume $\text{Re}[\alpha_{\text{tip}}] = \text{const}$. The same would work for multiple protrusions on the tip with $\text{Re}[\alpha_{\text{sample}}] = \text{const}$ and all the measured variation coming from $\text{Re}[\alpha_{\text{tip}}]$ ($\text{Re}[\alpha_{\text{tip}}]$ being different for each bump on the tip). Fig Ab potentially supports this interpretation, as it seems to show a lot of very similar bumps on the sample (and possibly being a multiple tip image?).

Also, different tips (A and C) would then lead to different geometry of bumps. Have you ever imaged the same NP with 2 different tips?

3. Tip effects:

'It is not necessary to be concerned about the possibility that tip features may appear in the image because ghost features do not appear in the PiFM image unless the tip contacts the sample.'

Please provide the error signal from the feedback – would experimentally demonstrate that no sample contact was made.

Where does the asymmetry come from in referee response Fig B c, d and f? The sketch b) is axially symmetric with respect to $x=0$, f) is completely asymmetric???

I think a more careful discussion of resolution is warranted, as the main aim of the paper is to demonstrate nanometer scale E-field resolution in 3D.

////////////////////////////////////

List of changes

////////////////////////////////////

We have made following revisions for the better explanation of our work and the readability of the manuscript.

1) To answer the reviewer's comment1-1, we have improved the three-dimensional data of the photoinduced force in the Fig. 4 in the main text. In addition, we have modified the sentences for the explanation for Fig. 4 in p.3, right column, line 84 to line103. Also, we have added the same figure for the different wavelength of light in Supplementary Information as Fig. S13.

2) To answer the reviewer's comment2-1, we have modified the sentence in p4, left column, line 8 to line 16.

3) To answer the reviewer's comment3-1 and 3-2, we have added the sentences in the main text (p.2 right column, line 68 to 78 and p.4 in the method section, left column, line 36 to 39).

4) To answer the reviewer's comment4-1, we have added the section for the explanation of the feedback error of the high-resolution imaging (in p.7 in the Supplementary Information).

5) To answer the reviewer's comment4-2, we have improved the theoretical results and explanation for the high-resolution imaging in p.14 to p15, the section of the "Theoretical analysis resolution" in the Supplementary Information.

6) To answer the reviewer's comment4-3, we have replaced Fig. S1 in the Supplementary Information into the new TEM images and added some descriptions about the nano-scale structural features.

7) To answer the reviewer's comment4-4, we have modified the explanation for Fig. S14 in the Supplementary Information (which was Fig. S12 in the previous Supplementary Information.)

The corrected parts in the main text and the Supplementary Information are indicated as the red color, and also, for the readability, English expressions have been improved throughout the manuscript.

Reply to Reviewers

We would like to thank the reviewer for the critical, yet positive review of our manuscript, “Optical Force Mapping at the Single-Nanometre Scale.” The reviewer comments and suggestions have strengthened our manuscript. In particular, reviewer’s advice that we should provide more detailed discussions to dispel the concerns about the possibility of the artifact generated by the contact potential difference, feedback control, and so on is quite important and useful to substantially improve the quality of our manuscript. We have attempted to respond to all the comments and incorporate all the suggestions made by the reviewer. The reviewer comments are followed by our responses, and modifications/revisions made in the main text. We hope that the revised manuscript will be suitable for publication in *Nature Communications*.

Reviewer’s comment 1-1

Fig 4 in the manuscript shows the main result. This figure is very hard to read – what are the gray scales and color scales in 4b? Contrast of non-blue arrows in 4a very weak. Given that the blue short arrows in 4a are essentially background – why plot them? If this paper is to be published the major aim of it (3D mapping of optically induced fields) needs to be presented much better.

I would have loved to see a 3D field as a result of different wavelengths of illumination – maybe even a difference map? I think the stated aim of the paper has been demonstrated – but it needs to be presented much better. The measurements also open many questions, which should either be discussed or mentioned for future work.

Our reply 1-1

We thank the reviewer for pointing out the obvious errors and missing information from Fig. 4, and apologise for the confusion. As pointed by the reviewer, this figure includes significant information that demonstrates the potential of our method. We have substantially improved Fig. 4 for better visualisation by separating the panels for the forces to the horizontal direction ($x - y$) and vertical direction (z). We now demonstrate the strength and direction of the forces by colours and arrows, Fig. R1, respectively. Further, we provide separate panels for different Δz . The forces to the horizontal direction ($x - y$) are shown in a, c, e). Those of vertical direction (z) are depicted in b, d, f). a-b), c-d), and e-f) are for $\Delta z = 2.0, 1.0,$ and 0.0 nm, respectively. This figure now demonstrates how the 3D force vectors change with the change of Δz .

Figure R1: Improved 3D mapping of the photoinduced force using 660 nm. a, c, e) Photoinduced force of F_x, y_{pif} . b, d, f) Photoinduced force of F_z . a-b), c-d), and e-f) $\Delta z = 2.0, 1.0,$ and 0.0 nm, respectively. g, h) Calculated photoinduced force of F_x, y_{pif} and F_z at $z = 1$ nm.

We are grateful to the reviewer for suggesting to show 3D maps of different wavelengths. We agree that the images of 3D field by different wavelengths more clearly demonstrate the powerful aspect of our method. We have incorporated the 3D force mapping with 520 nm wavelength of Fig. R2 as Fig. S13 in the Supplementary Information. Although the wavelength difference is not very clear as compared with the PiFM images in the main text, we consider this information is useful for readers to know that our method can acquire different images of the same sample with the different wavelengths.

Figure R2: 3D photoinduced force map using 520 nm wavelength. a, c, e) Photoinduced force of F_x, y_{pif} . b, d, f) Photoinduced force of $F_{z_{pif}}$. a-b), c-d), and e-f) are for $\Delta z = 2, 1,$ and 0 nm, respectively.

Reviewer's comment 2-1

Nanometer scale imaging of light induced electrical fields is important to understand many phenomena – catalysis, biological processes etc. As such, a demonstration in UHV is a good first step – I would like to see more of a discussion of how the technique presented will be transferrable to liquids or gas, given the well-known problems

of PiFM contrast.

What I find worrisome in this context is the fact that the measured signal is 100,000 times smaller than the expected PiF signal. Plausibility arguments are made for why one is so far off - but this should be more clearly stated in the paper. Some of the assumptions could be tested by careful experiments (and should be done down the road). This small signal does not bode well for transferring this technique to liquids or gas, where AFM sensitivity is drastically reduced. It also leads to a large sensitivity to other sources of PiF 'signals' not directly related to electrical fields. This discrepancy should be stated more clearly as a challenge.

Our reply 2-1

We thank the reviewer for the critical remark. The magnitude of the detected photoinduced force shown in the manuscript is about ~ 10 pN with the decay length of ~ 1 nm, as can be seen in Fig. 3b in the main text. Thus, the spring constant of the gradient force is ~ 10 pN/nm. On the other hand, in the previous report, "Phys. Rev. B 94, 195407 (2016)" for example, the spring constant of the photoinduced force is also ~ 10 pN with the decay length of ~ 1 nm. Therefore, we deduced that our detected signal has a reasonable strength and not very small compared with the other reports.

The force sensitivity in ambient and liquid conditions becomes one hundredth of that in UHV. However, it is easy to increase the laser power under the tip over 100 times by using variety of lenses including objective lens with large NA to focus the laser light for the measurement in ambient and liquid conditions. In addition, the measured photoinduced force in our experiment is ~ 10 times larger than the theoretical expectation, (see Supplementary Information, the section of "Magnitude of photoinduced force", p.16, line 1 to line 20. The force directing z in our measurement is 9600 times larger than the theoretical calculation. If we consider the experimental condition in detail for theoretical calculation, we can estimate 600×3 times larger theoretical force.) though the measured condition is not complete (small NA of focusing lens). Therefore, the PiFM measurements in liquids or gaseous conditions will be our target in near future.

We consider that the information on availability of PiFM in ambient and liquid conditions is useful for readers, and hence, we have added a comment on the above issue (in the main text, p.4, left column, line 8 to line 16). As a side note, in our Supplementary Information, we discuss why the theoretically obtained force value by DDA simulation is 9600 times smaller than that for the vertical direction obtained in the experiment.

Reviewer's comment 2-2

If one wanted to be really critical, one could state that contrast on the 1nm level is observed due to light shining on a sample, the origin of which is unclear, but possibly due to PiF. This however would be more of a criticism of the current understanding of contrast in the PiFM field as a whole than the original and interesting results presented in this paper.

Our reply 2-2

As this comment is related to your next comment, we would like to elaborate on this comment in our next reply.

Reviewer's comment 3-1

I am not convinced that light induced contact potential differences do not play a role – why is NO surface photovoltage (SPV) observed is very puzzling. This group has great experience in performing KPFM – I would only accept to publish this paper if I saw KPFM data on nanorods with/without light. Why not just show data and settle this discussion once and for all? Also, CPD data would allow a better determination of the minimal separation between tip and sample – crucial for the generation of 3D plots of the E-field. The volume of charge separation is not huge – in particular if one argues 1nm spatial resolution. . .

Potentially all the contrast comes from surface potential differences. Variations in SPV could explain the observed highly localized spots – traps being photo deactivated? The interpretation of the data would then change dramatically. In particular, comment 2 of referee 1: 'Surface potential: If not compensated locally, any variation translates into Δf used for topography feedback, hence height errors arise that will affect the magnitude of the gradient force detected. This would also affect the ratios in Fig. 1g,k.' is not answered satisfactory. The author response: 'Therefore, the images that we compared were measured completely at the same time, and hence, the tip-sample distance variation is entirely the same and the feedback control should not affect the comparison between the images for the different wavelengths even if surface potential and surface photovoltage on the QD exist. This is a strong point of our measurement technique.' This is only true if Au and the nanorod have the same CPV – highly unlikely! If different wavelengths activate traps or lead to spatially and temporally varying SPV on the nanorod then the deduced electric field is a convolution of several physical phenomena.

Figure S7 is not sufficient to demonstrate the absence of KPFM or SPV contrast – I want to see KPFM parabolas with and without light over the different regions as well as KPFM with/without light simultaneously to the data presented. I would in fact expect a difference in SPV!

Our reply 3-1 (and 2-2)

We thank the reviewer for giving us a chance to provide more in-depth discussion on the problem of the surface photovoltage (SPV). We agree that KPFM data on nanorods convinces readers that our PiFM image with less than 1 nm resolution is solely and exclusively due to the photoinduced force, and we should have shown corresponding data in the previous manuscript. Unfortunately, it is not easy to obtain clear KPFM signals in the case of our system because the tip becomes unstable when the voltage is applied. The several $\Delta f - V$ spectra measured on the edge of the QD are shown in Fig. R3. The obtained spectra are stable in the range of $-100 \sim 100$ mV, but above 100 mV (high voltage), the spectra rumple, indicating unstable tip. Hence, the KPFM signals could not be obtained in our experiment.

Figure R3: $\Delta f - V$ spectra at a site of the nanoellipsoid with the illumination.

However, we can obtain the SPV map, which provides the data that can be equivalent or provides direct evidence to show the absence of SPV. We elaborate this method as follows:

First, we conducted the frequency shift vs bias voltage spectroscopy, which does not require the feedback control of the applied voltage on QD rod, edge, and gold surface with and without light. The spectra with light show the same features of that without light, which means that the contact potential difference (CPD) are almost the same between both the cases. Here, we averaged the spectra which are obtained with illumination on each site to avoid the instability of tip shown in Fig. R3. Then, we compared them with data without illumination in Fig. R4.

The detected CPD difference between with and without light is ~ 10.9 mV and -1.25 mV on the rod and edge of the ZAIS QD, respectively. Here, we should note that these values are too small to detect in the PiFM measurement and within the range of the error bar.

Figure R4: $\Delta f - V$ spectra on the nanorod, nanoellipsoid, and gold surface with and without the illumination.

Further, we investigated whether the influence of the SPV by the light illumination is involved in PiFM images, based on the electrostatic force interaction between the tip and the surface. If SPV is generated by the light illumination with the modulation frequency on the surface, the electrostatic force, F_{ES} , between the tip and the surface is given by,

$$F_{ES} = \frac{1}{2} \frac{\partial C}{\partial z} (V_{DC} - V_{CPD} + V_{SPV} \cos \omega_m t)^2$$

$$= \frac{1}{2} \frac{\partial C}{\partial z} \left((V_{DC} - V_{CPD})^2 + 2(V_{DC} - V_{CPD})V_{SPV} \cos \omega_m t + V_{SPV}^2 \frac{1 + \cos 2\omega_m t}{2} \right) \quad (1)$$

Here, V_{DC} and V_{CPD} are the DC bias voltage and the CPD between the tip and the surface, respectively. V_{SPV} is the surface photovoltage on the surface. The ω_m component of the frequency shift $\Delta f(\omega_m)$ of the oscillating cantilever is given by,

$$\Delta f(\omega_m) = -\frac{f_0}{2ka_1} \int_{\phi}^{2\pi/\omega+\phi} (F_{ES}(z, \omega_m) + F_{pif}(z, \omega_m)) \cos(\omega t + \phi_1) dt$$

$$= -\frac{f_0}{2ka_1} \int_{\phi}^{2\pi/\omega+\phi} \frac{\partial C}{\partial z} \cos(\omega t + \phi_1) dt V_{SPV} (V_{DC} - V_{CPD}) \cos \omega_m t - \frac{f_0}{2ka_1} \int_{\phi}^{2\pi/\omega+\phi} F_{pif}(z) \cos(\omega t + \phi_1) dt \cos \omega_m t \quad (2)$$

Here, the first and second terms in the right hand side in Eq. (2) are related to SPV and the dipole-dipole interaction (namely, the photoinduced force), respectively. This equation indicates that if SPV is not generated ($V_{SPV}=0$), the modulated component of the frequency shift does not depend on V_{DC} and becomes constant (namely, only offsets appears). On the other hand, if SPV is generated ($V_{SPV} \neq 0$), $\Delta f(f_m)$ depends on V_{DC} , and the spectra of $\Delta f(f_m)$ vs. V becomes a tilted straight line. Figure. R5 shows the $\Delta f(f_m)$ as a function of V_{DC} measured on the nanorods and nanoellipsoids. In Fig. R5, we can see that the lines are not tilted though they are noisy, and only offsets were observed. That means in the spectra $V_{SPV} = 0$ and $F_{pif} \neq 0$ in Eq. (2).

Figure R5: Photoinduced force signal vs. bias voltage spectra ($\Delta f(f_m)X - V$). The sites where the spectra were obtained are on nanorod and nanoellipsoid. The wavelength of the laser light to induce the photoinduced force were 520 nm and 660 nm.

Moreover, we have evaluated the slope of $\Delta f(f_m)$ as a function of V_{DC} site by site and visualised them as 2D map, as shown in Fig. R6 (b). Here, the lowest value in colour scale of this image was set as the detectable limit of the CPD in the KPFM measurement that is 10 mV. The CPD signal is not detectable, unless the signal is larger than the lowest detectable limit. The detectable limit is determined by the following procedure. First, considering Eq. (2), $-\frac{f_0}{2ka_1} \int_{\phi}^{2\pi/\omega+\phi} \frac{\partial C}{\partial z} \cos(\omega t + \phi_1) dt$ was evaluated by the parabola fitting in the Fig. R4 as $\sim 110 \text{ Hz/V}^2$ at all sites. Therefore, the tilt of Eq. (2) is obtained by $-\frac{f_0}{2ka_1} \int_{\phi}^{2\pi/\omega+\phi} \frac{\partial C}{\partial z} \cos(\omega t + \phi_1) dt \times V_{SPV}$, and if the $V_{SPV} = 10 \text{ mV}$, the detectable limit of the tilt is determined as $\sim 110 \text{ Hz/V}^2 \times 10 \text{ mV} = 1.1 \text{ Hz/V}$. Thus, the Fig. R6(b) demonstrates that V_{SPV} is off-scale low at almost every point. Especially, all of the V_{SPV} on the positions of the nanorod, where we discussed in the main text, are completely off-scale. Although one pixel shows the larger value than the detectable limit, this is an apparent slope due to unstable tip at high bias voltage in the measurement, and the line profile in the image (b) does not show the small protrusions and even the edge feature of the ZAIS QD.

Figure R6: **a** Topographic image (Δz). **b** Tilt of $\Delta f(f_m)$ image. **c** Line profile of **b**

The above result definitely indicates that in PiFM, the dipole-dipole interaction between the tip and the sample is dominant and the influence of the SPV is negligible. The reason why SPV of the nanorods are undetectable are considered that the internal electric field in the QDs, which causes the charge separation, is very weak due to three dimensionally small size of the QDs, the SPV of the nanorods are undetectable. In addition, we would like to emphasise that the excitation light energies (660 and 785 nm) are lower than the bandgap of the QDs. This experimental condition strongly supports that the origin of the PiFM signal is not SPV.

We feel that the comments and questions by the reviewer about the issue of SPV are very valuable, and we agree that the discussions on this point will be very helpful for readers to understand the reliability of our demonstrated PiFM data. Therefore, in the revised manuscript, we have incorporated more elaborate arguments on the issue of the SPV in our PiFM measurements (p.2 right column, line 68 to 73 and p.4 in the method section, left column, line 36 to 39).

Reviewer's comment 3-2

In their answer, the authors state 'In the case of the bulk, the surface photovoltage variation in bulk semiconductors due to the separation charge region is $\sim 100nm$. However, this length is substantially longer than the QD size. Therefore, surface photovoltage should not generate on the surface of the QDs used in the present study.' This is correct for equilibrium situations. However, empirically, many groups have observed differences in KPFM with/without light in monolayers of organic molecules (where the argument above would apply; hence the request for empirical SPV data.

In passing, I observe that S7 Is used to argue (correctly, in my opinion) that 'the large difference in n indicates that the detected PiFM signal does not reflect the thermal expansion.' But - what proof is there that vdW dominates? How big is the change of CPD between Au and NP?

Our reply 3-2

We are grateful to the reviewer for posing a critical question. As elaborated in the previous reply, the CPD is almost the same between the cases with and without light illumination. Moreover, we confirmed that SPV does not affect the essence of our analysis of PiFM measurements. We consider that three dimensionally small size of the QDs, crucial for the internal electric field in the QDs, which causes the charge separation, is very weak. Further, we consider the excitation energy is lower than the bandgap energy of the QDs that should be taken into account to explain the observed results indicating negligible SPV. We have added a concise comment on this issue in the revised manuscript.

Reviewer's comment 4-1

I agree that one clearly observes some spatial variation of the signal on the order of 1nm. I think the authors should be more careful in arguing the origin of this resolution. Stating the open questions will stimulate more research.

Three different lines of argument for the 1nm resolution are considered by the authors in response to comment 3 of referee 1 on the resolution:

1. Cross talk: Referee 1 states 'Fiddling a bit with brightness and contrast of the corresponding topography scan (unfortunately at the end of the color scale), some of the features on the ellipsoid and rod also become discernible.' Figure A in the response seems convincing, but how can one exclude cross-talk between channels, e.g. due to feedback imperfections? Only the feedback signal (i.e. the black line in A) is shown. How big is the feedback error signal?

Our reply 4-1

We thank the reviewer for the insightful question. It can be seen in Fig. R7 that the feedback error is almost negligible and it cannot appear in PiFM signal as a cross-talk. Fig. R7a is the topographic image (Δz) of the edge of the ZAIS QD. Fig. R7b is the feedback signal image. This image was measured simultaneously with the PiFM image. The feedback signal is the frequency shift signal in FM-AFM defined as -20 Hz. The difference of Δf

from -20 Hz is the feedback error. Apparently, from this figure, we understand that the feedback error is almost negligible, and we do not worry about the effect of the cross-talk. (We can estimate the feedback error in our experiment as $\sim \pm 0.02$ Hz. The maximum and minimum value of the scale bar is due to the measurement noise.) We feel this information would be helpful for readers, and hence, we have incorporated it in the Supplementary Information.

Figure R7: (a) Topographic image (Δz) of the edge of the ZAIS QD. (b) Feedback signal image ($\Delta f = -20$ Hz). The difference of Δf from -20 Hz is the feedback error.

Reviewer's comment 4-2 (a)

0.5 nm is a close distance, typically used for tunneling, which is due to wave function overlap. So a classical model (as argued) does not apply.

Our reply 4-2 (a)

As pointed out by the reviewer, various quantum effects appear on the atomic scale, including quantum tunnelling and overlapping of the wave functions. It would be exciting to analyse the quantum effects in PiFM research, and we would work on it in future. However, in this study, we have analysed a phenomenon that can be described in terms of classical approximation. As pointed out by the reviewers, we should carefully consider the overlap of the wave functions due to the proximity between the probe and the sample. In order to safely handle the analysis under the classical approximation, we have assumed the tip-sample distances more than 4 Angstroms. This assumption is based on the earlier work [11], cited in Supplementary Information. In this report [11], the authors theoretically compared the classical and quantum analysis of plasmon generation in metal nanogaps and revealed that the classical and quantum analyses are consistent with each other when the gaps are more than 3.5 Angstroms apart. (ref. [11] in our Supplementary Information: R. Esteban, *et al.*, Faraday discussions, 178, 151 (2015).)

Reviewer's comment 4-2 (b)

Why should a gold tip support 1nm rough features (or 'atomic scale protrusions' ...)? Gold has a high diffusion constant – such a tip would not be stable, in particular when illuminated by light. I disagree with the authors statement 'As is well known, the tip of a gold-coated probe has many atomic-scale protrusions.' – all published TEM images of Au tips I am aware of show that they are smooth on a nm scale. References for the statement made? In the supplemental section 'The assumption of the existence of atomic scale protrusions on the tip is reasonable, and we consider that the actual situation is closer to that depicted by Fig. S12 than that depicted by Fig. S12b;' In fact I would argue that the tip in Fig S12b is more realistic. Give me references for why is this a 'reasonable' argument! (The answer cannot be that the modelling supports the experiment, as this is a circular argument not supported by independent evidence).

Our reply 4-2 (b)

We would like to thank the reviewer for allowing us to reconsider the shape of the tip apex. Considering the reviewer's comment, we have attempted additional modelling for the theoretical calculation by referring to the reported SEM image from Fig. 4(a) in ACS Photonics 2018, 5, 2, 390-397. The nanometre-scale structures are observed, even at room temperature, as shown in the tip SEM image. (A scale bar of 5 nm is added by us for the eye guide.) From the image, we find that the surface of the gold-coated tip has protrusions with the size around 4 ~ 5 nm.

Figure R8: SEM image of the gold coated tip in ACS Photonics 2018, 5, 2, 390-397.

Fig. R9 shows our calculation results of photoinduced force assuming such structures on the gold-coated tip. These results confirm that even nanocavities enables sufficient resolution, which supports our claim for the PiFM resolution. In our model, a structure with a diameter of 4 nm is somewhat buried in the mother body of the tip. Note that how the structure is buried does not affect the essence of the result.

Figure R9: (a) Schematic model of the tip with protrusions (nanocavity). (b) The line profiles of the photoinduced force. The red and blue lines are at tip-sample distance $d = 0.4$ and 0.5 nm, respectively. (c) The line profile of the gradient of the photoinduced force.

Reviewer's comment 4-3

It is stated that the contrast is dominated by the front tip protrusion. In contrast to the model, the sample does not seem to show topographical variation. What then is the physical origin of the spatial variation of properties leading to the stated ~ 1 nm spatial resolution? This is also where local KPFM contrast (=material contrast) would provide corroborating evidence.

Our reply

We are grateful to the reviewer for this question, which highlights the vital point of our result. Figs. R10 shows the TEM images of the ZAIS QDs synthesised by our group. As we can see in Fig. R10b, there are some topographical protrusions with 1 nm scale on the surface of the QD. In the synthesis process of the ZAIS QD, the QD surface is not completely flat and showcases some roughness. Such roughness was visualised in the TEM image, as reported in the previous study (Fig. 3(b) of "J. Phys. Chem. C 2018, 122, 25, 13705–13715"). We assume that the origin of the spatial variation with 1 nm scale in PiFM images is due to the topographical protrusions on the surface of the QD as modelled in Fig.R9.

Figure R10: TEM images of ZAIS QD.

Even though such nanocorrugations were often observed in the TEM images, the material and the property of the corrugation have not been elucidated by the current technologies. Therefore, the results of the PiFM measurements, which is a kind of optical spectroscopy, are scientifically significant to shed light on such unknown regions. We will incorporate the relevant statement in the main text and change the TEM image in the Supplementary Information into Fig. R10. As the dominant forces are different between the AFM and PiFM, the topographical protrusions were not visualised in AFM image, Fig. 2a in the main text, despite having 1 nm PiFM image resolution. The dominant force visualised in the PiFM image is photoinduced dipole-dipole interaction, whereas those visualised in the AFM image are van der Waals force and electrostatic force. The forces detected in an AFM have longer decay length than the photoinduced dipole-dipole force. In fact, the decay length of Δf for the former is ~ 2 nm, which is over three times longer than the decay length of photoinduced force signal ($\Delta f(f_m)$) as shown in Figs. S7a and b in Supplementary Information.

Further, in the tilt image (that we visualised as the SPV image in Fig. R6), there is no nanoscale feature. (That nanoscale feature is visualised in Fig. 2 in detail). Therefore, we conclude that the observed nanoscale feature is not originated from SPV but from the optical force.

Reviewer's comment 4-4

In the Suppl. Section the authors state 'Considering such a situation, as seen in Fig. S12a, we assume that many other protrusions exist whose heights are very similar to that of the central protrusion (the difference is only 0.2 nm in height). However, no ghost appears, and the central protrusion alone is sensitive to protrusions on the sample. I think this is potentially a good test. If the contrast essentially disappears for a 0.2nm distance change - how does the contrast in S12a change for a change of 0.2 nm in distance? · How does it change if the tip-sample distance is changed by 0.2nm in the model? · Does the contrast experimentally disappear for a change of 0.2nm in tip-sample separation? · What if the feedback error is 0.2nm?

Our reply 4-4

We thank the reviewer for pointing out the lack of picocavity explanation in the Supplementary Information. We do not claim that moving the tip 0.2 nm away will make the PiFM signal disappear. The presence of atomic-level protrusions between the metal nanogaps generates a remarkably localised enhanced electric field in the vicinity of the protrusions. In Fig R11 we show the electric field intensity in the vicinity of the picocavity. To prove that the significant localisation and enhancement of the electric field is due to the picocavity, we considered the case where the sample rods have no protrusions.

Figure R11: (a) The electric field intensity map. (b) The line profile of the electric field intensity at just under the tip apex.

Fig R11(b) is the line profile of the electric field intensity at just under the tip. From this figure, we understand that even though the difference in height between the longest protrusion and its neighbouring protrusions is as small as 0.2 nm, the difference in intensity of the localised electric field is very large for different protrusion sites. This is the reason why a high resolution is achieved, and no ghost image is observed. Further, we should note that PiFM detects gradient of the force, and we evaluated the forces at $z = 0.4$ nm and 0.5 nm in the case of Fig. R5 (b). A similar localised cavity effect is realised in the case of the simulation indicated in Fig. R9.

The feedback error was much smaller than 0.2 nm in our experiment. From Fig. R7b, we can estimate the feedback error in our experiment as ~ 0.02 Hz. Considering the value and the decay length of the tip-sample control signal (Δf) from Fig. S7 in Supplementary Information, this value changes the tip-sample distance as error by ~ 1.48 pm. This error is over 100 times smaller than the difference in height between the protrusions on the tip,

and such small difference can not affect the results.

In the revised Supplementary Information, we have added a concise discussion on this issue (in the Supplementary Information, the section of "Feedback error of the high resolution imaging", p.7, line 1 to line6).

Reviewer's Comment 4-5

The authors argue with a variation of $\text{Re}[\alpha_{\text{sample}}]$, and assume $\text{Re}[\alpha_{\text{tip}}] = \text{const}$. The same would work for multiple protrusions on the tip with $\text{Re}[\alpha_{\text{sample}}] = \text{const}$ and all the measured variation coming from $\text{Re}[\alpha_{\text{tip}}]$ ($\text{Re}[\alpha_{\text{tip}}]$ being different for each bump on the tip). Fig Ab potentially supports this interpretation, as it seems to show a lot of very similar bumps on the sample (and possibly being a multiple tip image?).

Our reply 4-5

As explained in the previous reply ('Our reply 4-4'), the appearance of a multiple tip image is not concerning because of the highly enhanced electric field localised at the longest protrusion, explained by Fig. R11.

Reviewer's Comment 4-6 Also, different tips (A and C) would then lead to different geometry of bumps. Have you ever imaged the same NP with 2 different tips?

Our reply 4-6

We thank the reviewer for this suggestion. We have not observed the same particle using different tips because this is a difficult measurement with the present technology (two tip PiFM measurement). However, such observation will provide significant information, and hence, we would like to develop a scheme for such measurement for a future study.

Reviewer's Comment 5-1

'It is not necessary to be concerned about the possibility that tip features may appear in the image because ghost features do not appear in the PiFM image unless the tip contacts the sample.' Please provide the error signal from the feedback – would experimentally demonstrate that no sample contact was made.

Our reply 5-1

As we discussed in 'Our reply 4-1', Fig.R7(b) is the feedback signal image. This image was measured simultaneously with the PiFM image. The feedback signal is the frequency shift signal in FM-AFM defined as -20 Hz. The difference of Δf from -20 Hz is the feedback error. We can estimate the feedback error in our experiment as ~ 0.02 Hz corresponding to the error by ~ 1.48 pm in the tip-sample distance. Thus, the feedback error is almost negligible and the possibility of sample contact is minimal.

Reviewer's Comment 5-2

Where does the asymmetry come from in referee response Fig B c, d and f? The sketch b) is axially symmetric with respect to $x=0$, f) is completely asymmetric???

Our reply 5-2

The origin of the asymmetry is due to the direction of the incident light. The light is obliquely incident on the quantum dot sample. The direction of incident light is illustrated in Fig. 1 of the main text.

REVIEWER COMMENTS

Reviewer #4 (Remarks to the Author):

I appreciate the detailed and thoughtful responses of the authors. I agree 100% with most of the responses. If the suggestion below is accepted by the authors I think this will be a stimulating and thorough publication breaking new ground in the field of PiFM.

I will recommend publication of this manuscript if the data presented in the response to question R-1 (concerning potential surface potential cross talk) is presented in the supplemental section. In particular, I would like to see Figs. R3 and R4 in the supplemental section. This is because I think the author's interpretation might be correct, but based on the data I cannot exclude that the authors are seeing a SPV artefact. Furthermore, I would like the authors to set the standard that KPFM data is presented simultaneously – I am convinced that many future researchers might otherwise fall prey to a KPFM 'artefact' that they interpret as PiFM.

Comment on responses:

Reply 3-1:

I do not understand why one can obtain SPV data (Fig R4) if one cannot obtain stable KPF spectra (Fig R3)? It is stated that the tip is unstable – but then why can one obtain R4?? What am I missing here? Is the instability observed in R3 due to charging/trap dynamics, the dynamics of which gets accelerated when shining light on the sample?

I understand that the difference in potential is below the PiFM detection limit. However, does this SPV difference lead to a change in tip-sample separation if not compensated? Asked differently: a change of 1 Hz correspond to more than 0.2nm distance change (based on Fig. S7b). So it could be a SPV effect cannot be completely excluded, in my opinion.

Some details concerning the figures R3 and R4:

Fig. R3: what do the different colors mean? What about the Au surface?

I don't understand the labeling (rod, edge) compared to the caption (nanorod, nanoellipsoid, gold). In R3 spectra with illumination of the nanoellipsoid are shown.

Is the nanoellipsoid data not shown in R4?

Were the spectra in R3 and R4 taken at the same tip-sample separation?

The frequency max in R3 is -10Hz, in R4 it is -16Hz. How big a tip-sample separation difference does this correspond to? My estimation (see also reply to Question 4-4 below) is 6Hz ~ 500pm, which is a substantial effect. How big is the tip-sample distance dependence of the KPFM or SVP spectra?

R4: Does this model/equation correctly describe the data? To convince me I want to see the residuals of a parabolic fit to the data. The reason for this is that if there are free charges in the system one would observe a deviation from this model.

Fig R5: show me a linear fit and the residuals of the fit. I think the lines are tilted for $V > 0.1V$

Supplemental Material:

Figure S1 and text:

Replace 'narrow-range' with 'field of view'

Alternative text for 'narrow-range': low and high magnification images

Signed:

Peter Grutter

Response to Reviewer

We would like to thank the reviewer for the thorough review of our manuscript, “Optical Force Mapping at the Single-Nanometre Scale.” We agree with the reviewer’s comment that PiFM data should be associated with KPFM data to exclude the possibility of observing artefacts by SPV. The reviewer’s suggestion that we should provide Figs. R3 and R4 and relevant explanations in the supplementary information is very helpful for this purpose. Although we cannot provide the data of the standard KPFM measurement in the present experiment, we have made our best effort to convince the readers by showing alternative data to prove the absence of SPV. We have also revised the Supplementary Information according to the reviewer’s suggestions. Our point-by-point response to the reviewer’s comments are included below.

We hope that the revised manuscript is now suitable for publication in *Nature Communications*.

Reviewer’s comment 1-1

I do not understand why one can obtain SPV data (Fig R4) if one cannot obtain stable KPF spectra (Fig R3)? It is stated that the tip is unstable - but then why can one obtain R4?? What am I missing here? Is the instability observed in R3 due to charging/trap dynamics, the dynamics of which gets accelerated when shining light on the sample?

Our reply 1-1

We apologize that the explanation for the data in the previous response letter was inadequate. We recorded many spectra through multiple trials, and the three curves in Fig. 3 are among those for the same position. On the other hand, the spectra in Fig. 4 (with illumination) were obtained by averaging over many such trials on both the nanoellipsoid and nanorod, respectively. Therefore, the spectra of Fig. R4 are smooth, yet the respective single spectra before averaging are similar to those in Fig. R3. As the reviewer pointed, the instability is particularly observed when the sample is irradiated with light and when the tip-sample bias voltage is applied.

Reviewer’s comment 2-1

I understand that the difference in potential is below the PiFM detection limit. However, does this SPV difference lead to a change in tip-sample separation if not compensated? Asked differently: a change of 1 Hz correspond to more than 0.2nm distance change (based on Fig. S7b). So it could be a SPV effect cannot be completely excluded, in my opinion.

Our reply 2-1

We thank the reviewer for this question. Here, we would like the reviewer to note that the frequency shift induced by the electrostatic force in our experiment between cases with and without illumination is ~ 0.05 Hz. In this case, the tip-sample difference by the SPV is estimated at ~ 3.5 pm. To explain this in more detail: From the linear fitting in “Our reply 5-1” shown below, we can safely say that CPD is much less than 10 mV. If the $V_{SPV} \sim 10$ mV, $-\frac{f_0}{2ka_1} \int_{\phi}^{2\pi/\omega+\phi} \frac{1}{2} \frac{\partial C}{\partial z} dt$ from the last transformation in Eq. (1) in the previous our reply can be estimated at ~ 0.09 (Hz/V)/(10)(mV) = 9 Hz/V². Therefore, the DC component of frequency shifts induced by the electrostatic force is

estimated at $9 \text{ [Hz/V}^2] \times (V_{\text{DC}} - V_{\text{CPD}})^2 \text{ [V}^2]$ and $9 \text{ [Hz/V}^2] \times (V_{\text{DC}} - V_{\text{CPD}} - V_{\text{SPV}})^2 \text{ [V}^2]$ for the cases without and with illumination, respectively (see Eq. (11) in the Supplementary Information of the present submission). These values are estimated at $\sim 9 \text{ [Hz/V}^2] \times 0.3^2 \text{ [V}^2] = 0.81 \text{ [Hz]}$ and $\sim 9 \text{ [Hz/V}^2] \times 0.31^2 \text{ [V}^2] = 0.8649 \text{ Hz}$. Therefore, the difference is $\sim 0.05 \text{ Hz}$. From the force curve in Fig. S7b (in the Supplementary Information of the present submission), we see that the tip-sample difference of 1 nm makes $\sim 15 \text{ Hz}$ in Δf . Therefore, the tip-sample difference by the SPV is estimated at $\sim 0.05 \text{ (Hz)/(15Hz)} \times 1 \text{ (nm)} \sim 3.5 \text{ pm}$. This value is quite small. Further, if the difference of 3.5 pm existed, the induced PiFM signal by the difference ($\Delta f(f_m)X$) is $\sim 0.9 \times \exp(-3.5 \text{ (pm)}/400 \text{ (pm)}) = 0.892 \text{ (Hz)}$, because the decay length is 400 pm (see Fig. S7a). Comparing with the case without illumination ($\sim 0.9 \times \exp(-0 \text{ (pm)}/400 \text{ (pm)}) = 0.9 \text{ (Hz)}$), we know that the difference in $\Delta f(f_m)X$ is 0.008 Hz . This value is below the noise level in the detection (see Fig. R3 in this reply or Fig. S10 in the Supplementary Information of the present submission) and quite small compared to the total detected value at $\Delta z = 0 \text{ nm}$ of 0.9 Hz (see Fig. S7 in the Supplementary Information of the present submission). Therefore, the induced tip-sample difference by the SPV does not affect the PiFM signal.

On another note, in our measurement, the PiFM images using several wavelengths of incident light in Fig. 1 and Fig. 3 in the main text were observed simultaneously. In other words, Fig. 1 (e, f), (i, j), and Fig. 3 (b, c), respectively, were observed in one scan. Therefore, the tip-sample distance difference between these images is absolutely 0, and our observed signal variations in Figs. 1 and 3 in the main text are not due to the change in the tip-sample separation.

Reviewer’s comment 3-1

Some details concerning the figures R3 and R4: Fig. R3: what do the different colours mean? What about the Au surface? I don’t understand the labelling (rod, edge) compared to the caption (nanorod, nanoellipsoid, gold). In R3 spectra with illumination of the nanoellipsoid are shown. Is the nanoellipsoid data not shown in R4?

Our reply 3-1

We apologize for the unclear labelling in the figure. In the caption of our previous reply, the rod and edge referred to nanorod and nanoellipsoid, respectively. Fig. R3 shows the spectra on the nanoellipsoid. As explained in "Our reply 1-1", we coloured the curves for the different trials only for better visibility. These curves show that the appearance of the instability in spectra is different. We have added these $\Delta f - V$ spectra in the Supplementary Information along with the spectrum on the Au surface in the figure given below.

Figure R1: $\Delta f - V$ spectrum at a site on the gold surface with the illumination.

Reviewer’s comment 4-1

Were the spectra in R3 and R4 taken at the same tip-sample separation? The frequency max in R3 is -10Hz, in R4 it is -16Hz. How big a tip-sample separation difference does this correspond to? My estimation (see also reply to Question 4-4 below) is 6Hz = 500pm, which is a substantial effect. How big is the tip-sample distance dependence of the KPFM or SVP spectra?

Our reply 4-1

We thank the reviewer for these questions. Yes, we took the $\Delta f - V$ spectra in Figs. R3 and R4 at the same tip-sample separation. The reason why the frequency maximum is seemingly different between Figs. R3 and R4 above 100 mV is due to the tip instability. In the case of the displayed trials, the curves below 100 mV are almost the same between Figs. R3 and R4, whereas they become rough curves where the values of the frequency shift especially above 100 mV jitter in Fig. R3. Actually, in this measurement, first, the tip-sample separation was kept without an applied bias voltage ($V_{DC} = 0$). Then, we turned off the tip-sample distance control quickly. Finally, we swept the applied bias voltage from -100 mV to 500 mV while measuring Δf . Hence, the $\Delta f - V$ spectra were obtained in the same tip-sample separation.

Reviewer’s comment 4-2

R4: Does this model/equation correctly describe the data? To convince me I want to see the residuals of a parabolic fit to the data. The reason for this is that if there are free charges in the system one would observe a deviation from this model.

Our reply 4-2

We have shown the measured spectra and parabola fit in the figure below. The residual sum of squares for 256 points of the blue, red, black, and green plots are 2.44239, 2.10373, 1.61127, and 2.64433, respectively. The coefficients of determination (R^2) of the fittings are 0.9973, 0.9984, 0.9976 and 0.9970, respectively. These values mean that the Δf spectra with and without illumination can be explained by the parabolic curves. We have added this figure in the Supplementary Information.

Figure R2: $\Delta f - V$ spectra on the nanorod and nanoellipsoid with and without illumination. The green and black plots represent the spectra on the nanoellipsoid with and without illumination, respectively. Similarly, the red and blue plots represent the spectra on the nanorod with and without illumination, respectively.

Reviewer’s comment 5-1

Fig R5: show me a linear fit and the residuals of the fit. I think the lines are tilted for $V > 0.1V$

Our reply 5-1

The linear fit is shown in the following figure.

Figure R3: Photoinduced force signal vs. bias voltage spectra ($\Delta f(f_m)X - V$). The sites where the spectra were obtained are on the nanorod and nanoellipsoid. The wavelengths of the laser light to induce the photoinduced force were 520 nm and 660 nm.

The tilts of each spectrum are 0.09006 (red), -0.02874 (black), 0.04192 (green), and 0.00224 (blue), respectively. The tilts are quite small and might not be consistent with each other. The tilts of the spectra can be changed by the noise and the instability of the tip. The intercepts of the spectra are 0.40975, 0.29122, 0.17959, and 0.15439, respectively. For example, if the SPV changes the CPD by 10 mV, $\Delta f(f_m)X$ using 660 nm on the nanoellipsoid can change $0.09006(\text{Hz}/\text{V}) \times 10(\text{mV}) \sim 0.0009$ Hz. This value is negligible compared to the intercept of the spectrum (0.40975 Hz). Hence, the SPV effect is minor.

Further, we would like to emphasize that if the detected signal was due to SPV (electrostatic force), $\Delta f(f_m)X$ should cross zero around CPD (~ 300 mV) as discussed previous papers. (*Applied surface science*, 157(4), 263-268 (2000), *Japanese Journal of Applied Physics*, 46(8S), 5626 (2007).) The residual sum of squares of the blue, red, black, and green plots for 256 points are 1.14739, 0.59844, 0.53927, 1.69504, respectively. The spatial map of these tilts was included in Fig. R6b in the previous response letter. From this figure, we can confirm that we need not be concerned about the SPV effect during PiFM measurement.

REVIEWERS' COMMENTS

Reviewer #4 (Remarks to the Author):

Thank you for the comprehensive responses. Very nice work! I am happy with the manuscript now and recommend that it is published as is.

One small remark: as an answer to my question 4.2 I did not expect a number (such as the residual sum of squares indicated in the response). I was expecting a plot of (data-fit function) as $f(\text{bias voltage})$. Such a plot can show small deviations very easily, even if the residual sum of squares is very small.